# Quantifying chromosomal instability from intratumoral karyotype diversity using agent-based modeling and Bayesian inference

**Andrew R Lynch[1,2], Nicholas L Arp[1], Amber S Zhou[1,2], Beth A Weaver[1,2,3], Mark E Burkard[1,2,4]***

[1]Carbone Cancer Center, University of Wisconsin-Madison, Madison, United States; [2]McArdle Laboratory for Cancer Research, University of Wisconsin-Madison, Madison, United States; [3]Department of Cell and Regenerative Biology, University of Wisconsin, Madison, United States; [4]Division of Hematology Medical Oncology and Palliative Care, Department of Medicine University of Wisconsin, Madison, United States

**Abstract** Chromosomal instability (CIN)—persistent chromosome gain or loss through abnormal mitotic segregation—is a hallmark of cancer that drives aneuploidy. Intrinsic chromosome mis-segregation rate, a measure of CIN, can inform prognosis and is a promising biomarker for response to anti-microtubule agents. However, existing methodologies to measure this rate are labor intensive, indirect, and confounded by selection against aneuploid cells, which reduces observable diversity. We developed a framework to measure CIN, accounting for karyotype selection, using simulations with various levels of CIN and models of selection. To identify the model parameters that best fit karyotype data from single-cell sequencing, we used approximate Bayesian computation to infer mis-segregation rates and karyotype selection. Experimental validation confirmed the extensive chromosome mis-segregation rates caused by the chemotherapy paclitaxel (18.5 ± 0.5/division). Extending this approach to clinical samples revealed that inferred rates fell within direct observations of cancer cell lines. This work provides the necessary framework to quantify CIN in human tumors and develop it as a predictive biomarker.

***For correspondence:**
mburkard@wisc.edu

## Editor's evaluation

The authors have developed a framework to quantify rates of chromosomal instability (CIN) in human tumors by fitting karyotype distributions inferred from low-depth DNA-sequencing to in silico models of CIN with karyotype selection pressures, sweeping through parameter space. This is particularly useful for the development of biomarkers for CIN, which is associated with cancer metastasis and drug resistance.

## Introduction

Chromosomal instability (CIN) is characterized by persistent whole-chromosome gain and loss through mis-segregation during cell division. Genome instability is a hallmark of cancer (*Hanahan and Weinberg, 2011*) and one type, CIN, is the principal driver of aneuploidy, a feature of ~80% of solid tumors (*Hancock et al., 2004*; *Knouse et al., 2017*; *Weaver and Cleveland, 2006*). CIN potentiates tumorigenesis (*Foijer et al., 2017*; *Levine et al., 2017*; *Silk et al., 2013*) and associates with therapeutic

**eLife digest** DNA contains all the information that cells need to function. The DNA inside cells is housed in structures called chromosomes, and most healthy human cells contain 23 pairs. When a cell divides, all chromosomes are copied so that each new cell gets a complete set. However, sometimes the process of separating chromosomes is faulty, and new cells may get incorrect numbers of chromosomes during cell division. Cancer cells frequently exhibit this behavior, which is called chromosomal instability', or CIN.

Chromosomal instability affects many cancer cells with varying severity. In cancers with high chromosomal instability, the number of chromosomes may change almost every time the cells divide. These cancers are often the most aggressive and difficult to treat.

Scientists can estimate chromosomal instability by counting differences in the number of chromosomes across many cells. However, many cells that are missing chromosomes die, resulting in inaccurate measures of chromosomal instability. To find a solution to this problem, Lynch et al. counted chromosomes in human cells with different levels of chromosomal instability and created a computer model to work out the relationship between chromosomal instability and chromosome number.

The model could account for both living and dead cells, which gave more accurate results. Lynch et al. then confirmed the accuracy of their approach by using it on a group of cells treated with a chemotherapy drug that causes a known level of chromosomal instability. They also used existing data from breast and bowel cancer, which revealed that levels of chromosomal instability varied between one mistake per three to twenty cell divisions.

Lower levels of chromosomal instability can be linked to a better prognosis for cancer patients, but it currently cannot be measured reliably. These results may help to reveal the causes of chromosomal instability and the role it has in cancer. If this method is successfully applied to patient samples, it could also improve our ability to predict how each cancer will progress and may lead to better treatments.

resistance (*Ippolito et al., 2020*; *Lee et al., 2011*; *Lukow et al., 2020*; *Pavelka et al., 2010*), metastasis (*Bakhoum et al., 2018*) and poor survival outcomes (*Bakhoum et al., 2011*; *Denu et al., 2016*; *Jamal-Hanjani et al., 2017*). Thus, CIN is an important characteristic of cancer biology. Despite its importance, CIN has not emerged as a clinical biomarker, in part because it is challenging to quantify.

Although CIN has classically been characterized as binary—tumors either have it or not—recent evidence highlights the importance of the rate of chromosome mis-segregation and the specific aneuploidies it produces. For example, clinical outcomes partially depend on aneuploidy of specific chromosomes (*Davoli et al., 2013*; *Sheltzer et al., 2017*; *Vasudevan et al., 2020*). Further, higher levels of CIN suppress tumor growth when they surpass a critical threshold, thought to be due to lethal loss of essential genes and irregular expression due to imbalanced gene dosage (*Funk et al., 2021*; *Silk et al., 2013*; *Weaver and Cleveland, 2008*; *Zasadil et al., 2014*). Moreover, baseline CIN may predict chemotherapeutic response to paclitaxel (*Janssen et al., 2009*; *Swanton et al., 2009*) and is proposed to both promote detection by or evasion from the immune system (*Davoli et al., 2017*; *Santaguida et al., 2017*). No single or standardized analytically valid measure of CIN has emerged and this gap has precluded its clinical validation as a prognostic or predictive biomarker.

Prior measures of CIN use various means to compare levels in tumors or populations, but do not establish a standardized quantitative rate. These prior measures include histologic analysis of mitotic defects (*Bakhoum et al., 2011*; *Jin et al., 2020*), fluorescence in situ hybridization (FISH) with probes to detect individual chromosomes (*Thompson and Compton, 2008*), and gene-expression methodologies like CIN scores (*Carter et al., 2006*). While these methods are readily accessible, they have significant drawbacks for clinical application. FISH and mitotic visualization approaches are laborious. Direct visualization of mitotic defects to measure CIN is only possible in the most proliferative tumors where enough cells are captured in short-lived mitosis. FISH typically quantifies only a subset of chromosomes, which will be misleading if there is bias toward specific chromosome gains/losses (*Dumont et al., 2020*). While gene expression scores are proposed as indirect measures of CIN, they are not specific to CIN and correlate highly with proliferation and structural aneuploidy (*Carter et al., 2006*; *Sheltzer, 2013*).

Single-cell sequencing promises major advances in quantitative measures of CIN by displaying cell-cell variation for each chromosome across hundreds of cells (*Navin et al., 2011*; *Wang et al., 2014*). However, selection poses another complication. To date, single-cell analyses have identified surprisingly low cell-cell karyotype variation, even when mitotic errors are directly observed by microscopy (*Bolhaqueiro et al., 2019*; *Gao et al., 2016*; *Kim et al., 2018*; *Nelson et al., 2020*; *Wang et al., 2014*). These observations highlight the confounding role of selection against aneuploid karyotypes in measuring CIN in human tumors. Indeed, selection reduces karyotype variance in cancer cell populations that directly exhibit mitotic errors (*Gerstung et al., 2020*; *Ippolito et al., 2020*; *Lukow et al., 2020*). Here, we seek to overcome gap by modeling chromosomal instability and explicitly considering the evolutionary selection of aneuploid cells, to derive a quantitative measure.

We describe a quantitative framework to measure CIN by sampling population structure and cell-cell karyotypic variance in human tumors, accounting for selection on aneuploid karyotypes. We built our framework on the use of phylogenetic topology measures to quantify underlying evolutionary processes (*Mooers and Heard, 1997*); in this case to quantify CIN from both the diversity and the aneuploid phylogeny within a tumor. Using an agent-based model of CIN, we determine how distinct types and degrees of selective pressure shape the karyotype distribution and population structure of tumor cells at different rates of chromosome mis-segregation. We then use this in silico model as a foundation for parameter inference to provide a quantitative estimate of CIN as the numerical rate of chromosome mis-segregation per cell division. We apply this model to quantify CIN caused by the chemotherapeutic paclitaxel in culture. Next, using existing single-cell whole-genome sequencing data (scDNAseq), we measure CIN in cancer biopsy and organoid samples. As a whole, this work provides a framework to quantify CIN in human tumors, a first step toward developing CIN as a prognostic and predictive biomarker.

## Results

### A framework for modeling CIN and karyotype selection

To assess intratumoral CIN via cell-cell karyotype heterogeneity, we considered how selection on aneuploid karyotypes impacts observed

**Table 1.** Base chromosome-specific fitness scores for individual models.

| CHR ARM | Selection model | | |
| --- | --- | --- | --- |
| | Gene Abundance | Driver Density | Hybrid |
| 1p | 0.04780162 | –0.0024018 | 0.02269992 |
| 1q | 0.04340321 | 0.03244362 | 0.03792341 |
| 2p | 0.02733655 | 0.02935717 | 0.02834686 |
| 2q | 0.04244054 | 0.03943267 | 0.0409366 |
| 3p | 0.02310412 | 0.03289695 | 0.02800053 |
| 3q | 0.0299756 | 0.05416736 | 0.04207148 |
| 4p | 0.01238195 | 0.01784909 | 0.01511552 |
| 4q | 0.03181796 | 0.02901324 | 0.0304156 |
| 5p | 0.01178443 | 0.04281166 | 0.02729805 |
| 5q | 0.03787615 | 0.01949934 | 0.02868775 |
| 6p | 0.02557719 | 0.02398619 | 0.02478169 |
| 6q | 0.02554399 | 0.00011625 | 0.01283012 |
| 7p | 0.0179588 | 0.09889284 | 0.05842582 |
| 7q | 0.03231589 | 0.06933314 | 0.05082451 |
| 8p | 0.01591728 | 0.02769564 | 0.02180646 |
| 8q | 0.0254942 | 0.05861427 | 0.04205423 |
| 9p | 0.01301266 | –0.0012941 | 0.00585929 |
| 9q | 0.02572657 | 0.04702681 | 0.03637669 |
| 10 p | 0.0112201 | –0.0364218 | –0.0126008 |
| 10q | 0.02750253 | 0.01142688 | 0.01946471 |
| 11 p | 0.01961858 | 0.03818621 | 0.0289024 |
| 11q | 0.03629936 | 0.01898784 | 0.0276436 |
| 12 p | 0.0142575 | 0.0551551 | 0.0347063 |
| 12q | 0.03659812 | 0.06273786 | 0.04966799 |
| 13 p | 0 | 0 | 0 |
| 13q | 0.02333649 | –0.0101539 | 0.00659128 |
| 14 p | 1.66E-05 | 0 | 8.30E-06 |
| 14q | 0.03792594 | 0.02557439 | 0.03175016 |
| 15 p | 0 | 0 | 0 |
| 15q | 0.03701306 | 0.0206566 | 0.02883483 |
| 16 p | 0.02383442 | 0.04334736 | 0.03359089 |
| 16q | 0.01900446 | –0.0071444 | 0.00593005 |
| 17 p | 0.01548573 | –0.0085975 | 0.00344414 |
| 17q | 0.03553586 | 0.04363474 | 0.0395853 |
| 18 p | 0.00627396 | 0.00533697 | 0.00580547 |
| 18q | 0.01434049 | –0.0263632 | –0.0060113 |

*Table 1 continued on next page*

*Table 1 continued*

|  | Selection model |  |  |
|---|---|---|---|
| 19 p | 0.02159372 | 0.05371416 | 0.03765394 |
| 19q | 0.02813325 | 0.00550338 | 0.01681831 |
| 20 p | 0.0089628 | 0.04351025 | 0.02623653 |
| 20q | 0.01526996 | 0.04993593 | 0.03260295 |
| 21 p | 0.00232369 | 0 | 0.00116185 |
| 21q | 0.01233215 | –0.0033092 | 0.00451147 |
| 22 p | 0.00013278 | 0 | 6.64E-05 |
| 22q | 0.02297134 | –0.0051581 | 0.0089066 |
| Xp | 0.01555213 | 0 | 0.00777606 |
| Xp | 0.02499627 | 0 | 0.01249813 |

chromosomal heterogeneity within a tumor. By modeling fitness of aneuploid cells, we observe chromosomal variation in a population of surviving cells. The selective pressure of diverse and specific aneuploidies on human cells has not been, to our knowledge, directly measured. Therefore, we employ previously developed models of selection.

In models of CIN, fit karyotypes are selected while unfit aneuploid karyotypes are eliminated over time (*Ippolito et al., 2020*; *Ravichandran et al., 2018*; *Sheltzer et al., 2017*; *Vasudevan et al., 2020*). We use two previously proposed models of aneuploidy-associated cellular fitness, as well as hybrid and neutral selection models. The Gene Abundance model is based on the relatively low incidence of aneuploidy in normal tissues and assumes cellular fitness declines as the cell's karyotype diverges from a balanced euploid karyotype (*Sheltzer and Amon, 2011*; *Zhu et al., 2012*). When an individual chromosome diverges from euploid balance (2 N, 3 N, 4 N, for example), its contribution to cellular fitness is weighted by its abundance of genes (*Figure 1—figure supplement 1A*, left). Alternatively, the Driver Density model assumes that each chromosome's contribution to cellular fitness is weighted by its ratio of Tumor suppressor genes, Oncogenes, and Essential genes (TOEs)(*Davoli et al., 2013*; *Laughney et al., 2015*). For example, Driver Density selection will favor loss of chromosomes with many tumor suppressors and favor gain of chromosomes replete with oncogenes and essential genes (*Figure 1—figure supplement 1A*, right). The hybrid averaged model accounts for both karyotypic balance and TOE densities (*Figure 1—figure supplement 1A*, middle). Using these fitness models, we assigned chromosome scores to reflect each chromosome's value to cellular fitness (*Figure 1—figure supplement 1B*, *Table 1*), the sum of which represent the total fitness value for the cell, relative to a value of 1 for a euploid cell. Further, we scaled the impact of cell fitness with a scaling factor, S, ranging from 0 (no selection) to 100 (high selection). While these models are approximations, they are nevertheless useful to estimate how mis-segregation and selective pressure cooperate to mold karyotypes in the cell population.

We employed these selection models in an agent-based model of exponential population growth wherein each cell has its own karyotype (*Figure 1* and *Figure 1—figure supplement 1*). Briefly, simulations started with 100 euploid cells and were run in discrete time steps with variable rates of selective pressure, S, and rates of chromosome mis-segregation ($P_{misseg}$, see definitions in *Table 2*). The rate—or probability—of mis-segregation events, $P_{misseg}$, is the measure of CIN. During each time step, cells have a $P_{division}$ ( = 0.5 for euploid) chance of dividing. Each dividing cell has a $P_{misseg}$ chance of improper segregation of each chromosome. Segmental chromosome breaks occur with a probability $P_{break}$, set at 0 or 0.5. After division, fitness (F) of each daughter is assessed. Cells are removed from the population if any given chromosome has copy number 0 or >6. The $P_{division}$ value of the remaining viable cells is adjusted by the cell's fitness under selection ($F^S$). Due to computational limitations, pseudo-Moran or Wright-Fisher models are employed to limit the modeled cell population (*Figure 1—figure supplement 1C, D*). These limits did not significantly affect the measures extracted from these populations (*Figure 1—figure supplement 2*). Thus, these models simulate an evolving population of aneuploid cells under given rates of CIN, $P_{misseg}$, and models and strength of selection.

## Evolutionary dynamics is imparted by CIN

To understand the interplay between CIN and selection, we simulated 100 steps of cell growth with CIN under each selection model. We varied the rate of CIN ($P_{misseg,c} \in [0, 0.001… 0.05]$ per chromosome; or 0–2.3 chromosome mis-segregations per division) and selective pressure ranging from none to heavy selection (S $\in [0, 2… 100]$). As expected, the simulated cell number increases rapidly to the pseudo-Moran cap of 3000, where it remains (*Figure 2A*). As displayed in *Figure 2B*, diversity of the

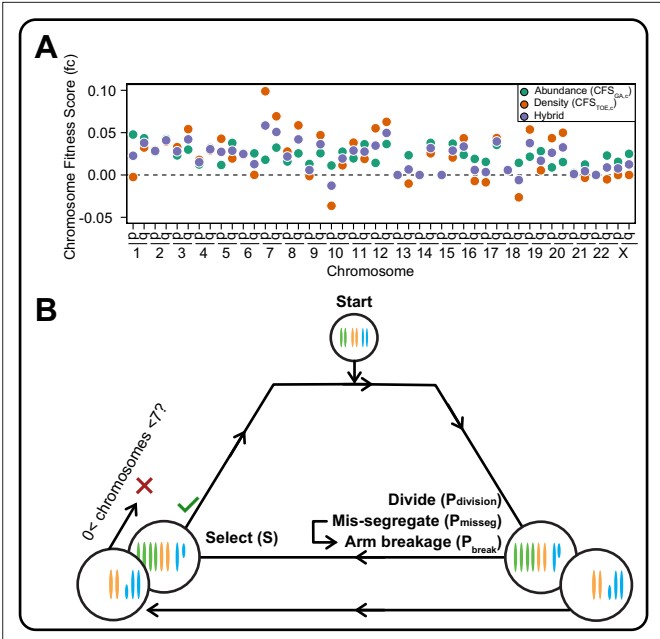

**Figure 1.** A framework for modeling CIN and karyotype selection. (**A**) Chromosome arm scores for each model of karyotype selection. Gene Abundance scores are derived from the number of genes per chromosome arm normalized to the number of all genes. Chromosome arms 13 p and 15 p did not have an abundance score and were set to 0. Driver Density scores come from the pan-cancer chromosome arm scores derived in **Davoli et al., 2013**, and normalized to the sum of chromosome arm scores for chromosomes 1-22,X. Chromosome arms 13 p, 14 p, 15 p, 21 p, 22 p, and chromosome X did not have driver scores and were set to 0. Hybrid model scores are set to the average of the Driver and Abundance models. The neutral model (not displayed) is performed with all cell's fitness constitutively equal to 1 regardless of karyotype. (**B**) Framework for the simulation of and selection on cellular populations with CIN. Cells divide (Pdivision starts at 0.5 in the exponential pseudo-Moran model and is constitutively equal to 1 for the constant Wright-Fisher model) and probabilistically mis-segregate chromosomes (Pmisseg ∈ [0, 0.001… 0.05]). After, cells experience selection under one of the selection models, altering cellular fitness and the probability (Pdivision) a cell will divide again (green check). Additionally, cells wherein the copy number of any chromosome falls to zero or surpasses 6 are removed (red x). After this, the cycle repeats. See Materials and methods for further details.

The online version of this article includes the following figure supplement(s) for figure 1:

**Figure supplement 1.** Expanded model of chromosome mis-segregation and karyotypic selection.

**Figure supplement 2.** Population growth limits do not bias population measures.

cell population, expressed as mean karyotypic variance increases over time, but also depends on mis-segregation rate, and selection levels (*Figure 2B*). As expected, high mis-segregation rates (P_misseg, Y axis) and low selection (S = 0; top row) enhance the variance of the population. Further, without selection (S = 0; top row) all models returned comparable profiles over time, resembling neutral selection. However, when selective pressure is applied (S > 0), the distinct profiles appear. The abundance model (first column) negatively selects against all aneuploid karyotypes and yields low heterogeneity that increases modestly with mis-segregation rate. With the Driver model (second column), there is a sharp increase in heterogeneity even at low mis-segregation rates, as this model favors specific aneuploid states that maximizes oncogenes and minimizes tumor suppressors. The Hybrid model falls between the other two. Results were not specific to the pseudo-Moran process of capping at 3000 cells—dynamics were similar in

**Table 2.** Parameters varied during agent-based modeling.

| Parameter | Description |
| --- | --- |
| Pmisseg | Probability of mis-segregation per chromosome per division |
| Pbreak | Probability of chromosome breakage after mis-segregation |
| Pdivision | Probability of cellular division per time step |
| S | Magnitude of selective pressure on aneuploid karyotypes |

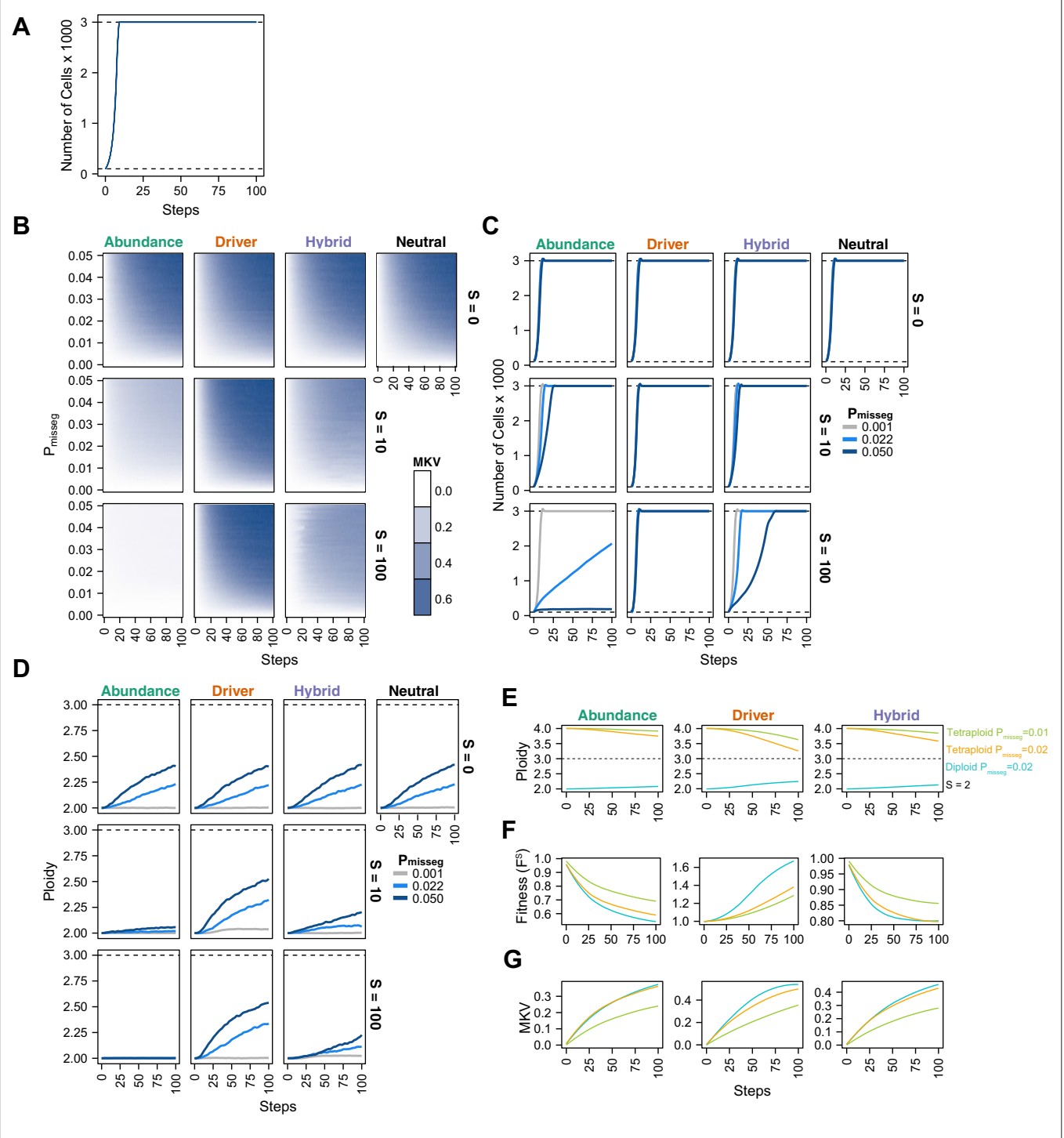

**Figure 2.** Evolutionary dynamics imparted by CIN. (**A**) Population growth curve in the absence of selective pressure ($P_{misseg}$ = 0.001, S = 0, n = 3 simulations). The steady state population in null selection conditions is 3000 cells. (**B**) Heatmaps depicting dynamics of karyotype diversity as a function of time (steps), mis-segregation rate ($P_{misseg}$), and selection (S) under each model of selection. Columns represent the same model; rows represent the same selection level. Mean karyotype diversity (MKV) is measured as the variance of each chromosome averaged across all chromosomes 1–22, and chromosome X. Low and high MKV are shown in white and blue respectively (n = 3 simulations for every combination of parameters). (**C**) Population growth under each model, varying $P_{misseg}$ and S. $P_{misseg} \in$ [0.001, 0.022, 0.050] translate to about 0.046, 1, and 2.3 mis-segregations per division respectively for diploid cells. (**D**) Dynamics of the average ploidy (total # chromosome arms / 46) of a population while varying $P_{misseg}$ and S. (**E**) Dynamics of ploidy under each model for diploid and tetraploid founding populations. $P_{misseg} \in$ [0.01, 0.02] translate to about 0.46 and 0.92 mis-segregations for diploid cells

*Figure 2 continued on next page*

*Figure 2 continued*

and 0.92 and 1.84 mis-segregations for tetraploid cells. (**F**) Fitness ($F^S$) over time for diploid and tetraploid founding populations evolved under each model. (**G**) Karyotype diversity dynamics for diploid and tetraploid founding populations. MKV is normalized to the mean ploidy of the population at each time step. Plotted lines in C-G are local regressions of n = 3 simulations.

The online version of this article includes the following figure supplement(s) for figure 2:

**Figure supplement 1.** Chromosomal instability and karyotype selection in constant-size populations approximating Wright-Fisher dynamics.

**Figure supplement 2.** Fitness of diploid and tetraploid CIN +populations.

the constant-population Wright-Fisher model (***Figure 2—figure supplement 1A, B***). These data illustrate how CIN and selection operate together to shape the karyotype diversity in the cell population.

High levels of selection against aneuploid cells are expected to impede cell growth. To visualize this, we quantified the population of viable cells with distinct models (***Figure 2C***). As expected with the Abundance model at S = 10 and S = 100, cells proliferated more slowly with higher rates of mis-segregation. By contrast, the Driver model saw no growth defect as they favored specific aneuploid states that are easily reached with missegregation. As before, the Hybrid model, is intermediate, and findings are not impacted by pseudo-Moran or Wright-Fisher restrictions on cell number (***Figure 2—figure supplement 1C***).

To further assess model dynamics, we examined time-course of average cellular ploidy—the number of chromosomes divided by 23. In many cases, the mean ploidy of the populations tend to increase over time (***Figure 2D***, ***Figure 2—figure supplement 1D***), particularly in the absence of selection (S = 0; top). This is likely due to a higher permissiveness to chromosome gains than losses in our model (since cells 'die' with nullisomy or any chromosome >6, the optimum is 3.0). With selection (S = 10; S = 100 rows), the models diverge. In the abundance model, populations remain near diploid. With the Driver model, the average ploidy increases more rapidly due to favoring aneuploidy states that favor high oncogenes and low tumor suppressors, consistent with previous computational models built on chromosome-specific driver densities (***Davoli et al., 2013***; ***Laughney et al., 2015***). Under the Hybrid model, ploidy increases modestly. Similar effects are seen with the constant-population Wright-Fisher model (***Figure 2—figure supplement 1D***). In sum, selection and mis-segregation cooperate to shape the aneuploid karyotypes diversity, cell proliferation and average ploidy in a population of cells, or a human tumor. Further, sampling karyotypes in a cell population does not allow direct determination of mis-segregation rates, as their diversity is influenced by other factors such as selective pressure, selection modality, and time.

In some tumors, genome doubling occurs early in tumor initiation relative to other copy number changes (***Bielski et al., 2018***; ***Gerstung et al., 2020***). Genome doubling is accomplished, for example, by endoreduplication, by failed cytokinesis, or by cell-cell fusion. Genome doubling buffers against loss of chromosomes and thereby favors aneuploidy. To determine how genome doubling impacts evolution in our model, we compared diploid and tetraploid founders (***Figure 2E–G***). Both diploids and tetraploids tend to converge toward the near-triploid state (ploidy ~3), as observed in many human cancers (***Carter et al., 2012***), although this is restrained to a degree with the Abundance and Hybrid models. Compared with diploid cells, tetraploidy buffered against the negative effects of cellular fitness in the Abundance model, despite generating similar levels of diversity over time (***Figure 2F and G***)— this is more pronounced when comparing $P_{misseg}$ = 0.1 in tetraploids versus $P_{misseg}$ = 0.2 in diploids to match the number of chromosome mis-segregations per division. This is consistent with the idea that tetraploidy serves as an intermediate enabling a near-triploid karyotype that is common in many cancers (***Bielski et al., 2018***; ***López et al., 2020***). By contrast, in the Driver model, tetraploidy did not provide a selective advantage to high-CIN tumors (***Figure 2F***). Similar fitness, karyotype diversities, and ploidy increases were obtained with a Wright-Fisher model of population growth (***Figure 2—figure supplement 1E-G***, ***Figure 2—figure supplement 2***).

Taken together, the agent-based model recapitulates expected key aspects of tumor evolution, lending credence to our model. Further, they illustrate the difficulty of inferring mis-segregation rates directly from assessing variation in karyotypes in human cancer. Nevertheless, this model provides a framework to incorporate selection to measure CIN through quantitative inference from the observed karyotypes, as we will demonstrate.

## Long-term karyotype diversity depends profoundly on selection modality

Some current measures of CIN are derived from karyotype diversity in the population. Yet, our model suggests that selection pressure will profoundly shape this diversity. To further understand the nature of karyotype diversity under selection, we evaluated their long-term dynamics, whether they exhibit clonality, and whether populations simulated under each model converge on a common karyotype.

We simulated diploid and tetraploid populations for 3000 time steps at a fixed mis-segregation rate, in an experimentally reported range, allowing for fragmentation of chromosome arms ($P_{misseg}$ = 0.003, $P_{break}$ = 0.5) (**Bakhoum et al., 2009**; **Bolhaqueiro et al., 2019**; **Weaver et al., 2007**) and S ∈ [1,25] (**Figure 3A**). We visualized copy-number heatmaps indicating karyotypes of sampled cells from the population. As expected, population diversity is limited under the Abundance model (**Figure 3B**). Even after 3000 time steps, only a small number of unique alterations and sub-clonal alterations ( + 13 p/–15 p/–22 p) existed, likely passenger alterations as they offer no fitness advantage in this model. Moreover, the karyotype average of 1500 cells across five replicates resembled a diploid karyotype (**Figure 3C**, row 1), indicating that the Abundance model provides stabilizing selection around the euploid karyotype. In fact, populations simulated under this model with elevated selection (S = 25) quickly reach a low, steady-state level of karyotype diversity and fitness while those with the unmodified selection values (S = 1) take a longer time to reach this steady-state and have similar levels of karyotype diversity and fitness as the other models (**Figure 3—figure supplement 1**). To identify any contingencies that may affect these associations, we performed the same simulation using several variants of our model. We found this steady state to be consistent for tetraploid cells as well as when we eased the upper ploidy constraint from $nc_c$ = 6 to an extreme $nc_c$ = 10, when we imposed a severe, 90% fitness reduction for all cells with a haploidy, and when we simulated populations under the Wright-Fisher model (**Figure 3C**, rows 2–4).

The Driver Density and Hybrid models generate much more diversity (**Figure 3B**) but nevertheless converge by 3,000 timesteps (**Figure 3—figure supplement 1**). Without selection (neutral model), there is high diversity and no convergence over time. Taken together, these demonstrate a high dependence on the model of selection. However, the models are not highly dependent on ploidy constraints, haploid penalties, or on selection of Pseudo-Moran or Wright-Fisher restriction of cell numbers. Taken together, long-term populations are strongly shaped by the model of karyotype selection for a given $P_{misseg}$, but relatively insensitive to other particular features of the model. This justifies our approach henceforth of varying only the selection model, the degree of selection (S), and $P_{misseg}$ to infer parameters from data via phylogenetic topology and Bayesian inference.

## Topological features of simulated phylogenies delineate CIN rate and karyotype selection

Given a model capable of recapitulating diversity and selective pressures, next we wish to infer $P_{misseg}$ as a measure of CIN from an observed population of cells. Phylogenetic trees provide insights into evolutionary processes of genetic diversification and selection. Moreover, the topology of the phylogenetic tree has been used as a quantitative measure of the underlying evolutionary processes (**Colijn and Plazzotta, 2018**; **Dayarian and Shraiman, 2014**; **Manceau et al., 2015**; **Neher et al., 2014**; **Scott et al., 2020**).

Here, chromosome mis-segregation gives rise to karyotype heterogeneity, and the population of cells is then shaped by selection. To evaluate this, we use chromosome copy number-based phylogenetic reconstruction, since mutation rates are not high enough in tumors to reliably infer cellular relationships, particularly with low-copy sequencing. Once phylogenies are reconstructed from simulated and experimental populations, the topological features phylogenies can be compared. These features include 'cherries'—two tips that share a direct ancestor—and 'pitchforks—a clade with three tips (**Figure 4A**). Additionally, we considered a broader metric of topology, the Colless index, which measures the imbalance or asymmetry of the entire tree. To understand how these measures are affected by selection in simulated populations, we reconstructed phylogenies from 300 random cells from each population simulated with a range of selective pressures taken at 60 time steps (~30 divisions under Hybrid selection; **Figure 4B**). As seen previously, aneuploidy and mean karyotypic variance (MKV) decrease with selective pressure, a trend that is robust at high mis-segregation rates (**Figure 4C**). By contrast, Colless indices increase with mis-segregation rates and selective pressures,

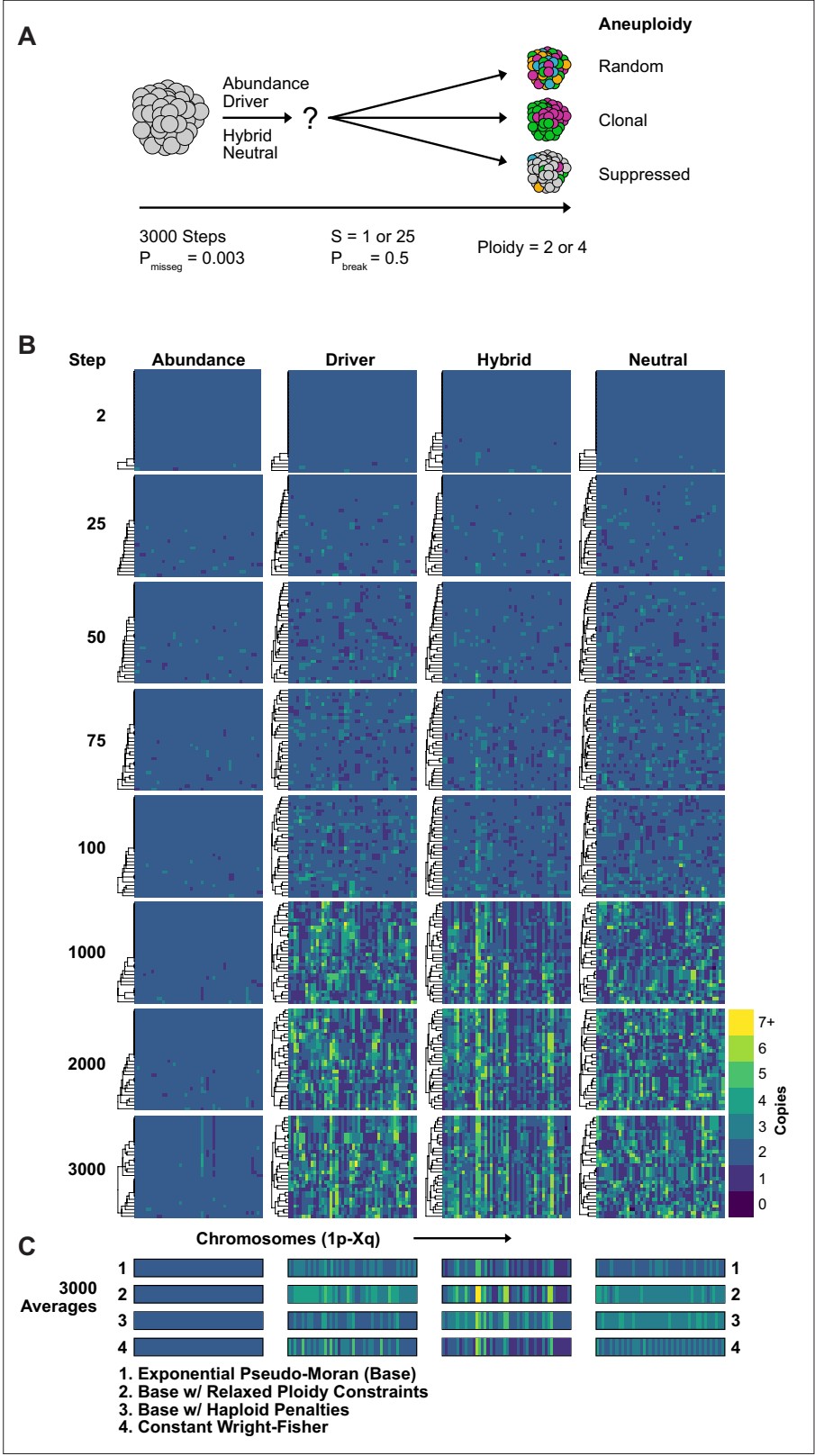

**Figure 3.** Karyotype diversity depends profoundly on selection modality. (**A**) Simulation scheme to assess long-term dynamics of karyotype evolution and karyotype convergence. (**B**) Heatmaps depicting the chromosome copy number profiles of a subset (n = 30 out of 300 sampled cells) of the simulated population with early CIN over time under each model of karyotype selection. (**C**) Average heatmaps (lower) show the average copy number across

*Figure 3 continued on next page*

*Figure 3 continued*

the 5 replicates for (1) the Exponential Psuedo-Moran (Base), (2) the base model with the upper copy number limit set to 10, (3) the base model that invokes a $F_M$ x 0.1 penalty for any cell with a haploid chromosome, (4) and the Constant Population-Size Wright-Fisher model. $P_{misseg}$ = 0.003; S = 25 (except Neutral model; S = 0); ploidy = 2.

The online version of this article includes the following figure supplement(s) for figure 3:

**Figure supplement 1.** Modeled population measures tracked over time.

---

as the resulting variation and selection generate phylogenetic asymmetry. Accordingly, this imbalance is apparent in phylogenetic reconstructions of simulated populations (***Figure 4D***). Cherries, by contrast, decrease with selection due to selection against many aneuploidies (***Figure 4C***). Pitchforks seemed less informative. Therefore, we tentatively selected 4 phylogenetic parameters that can retain information about chromosome missegregation—aneuploidy, MKV, Colless, and Cherries.

To characterize how well the four measures retain information about the simulation parameters, we performed dimensionality reduction with measures of karyotype heterogeneity alone (MKV and aneuploidy) alone and adding Colless and cherries—measures of phylogenetic topology (***Figure 4E***). This analysis indicates that when considering heterogeneity alone simulations performed under high CIN/high selection (yellow) and low CIN/low selection (red) associate closely, meaning these measures of heterogeneity are not sufficient to distinguish these disparate conditions (***Figure 4E***, left). These similarities arise because high selection can mask the heterogeneity expected from high CIN. By contrast, combining measures of heterogeneity with those of phylogenetic topology can discriminate between simulations with disparate levels of CIN and selection (***Figure 4E***, right). This provides further evidence that measures of heterogeneity alone are not sufficient to infer CIN due to the confounding effects of selection, particularly when the nature of selection is unclear or can vary. Together these results indicate that phylogenetic topology preserves information about underlying levels of selective pressure and rates of chromosome mis-segregation. Further, phylogenetic topology of single-cell populations may be a suitable way to correct for selective pressure when estimating the rate of chromosome mis-segregation from measures of karyotype diversity.

## Experimental chromosome mis-segregation measured by Bayesian inference

To experimentally validate quantitative measures of CIN, we generated a high rate of chromosome mis-segregation with a clinically relevant concentration of paclitaxel (Taxol) over 48 hr (***Figure 5A***). We treated CAL51 breast cancer cells with either a DMSO control or 20 nM paclitaxel, which generates widespread aneuploidy due to chromosome mis-segregation on multipolar mitotic spindles (***Zasadil et al., 2014***), verified in this experiment (***Figure 5—figure supplement 1A***). At 48 hr cells will have undergone 1–2 mitoses and, consistent with abnormal chromosome segregation, we observe broadened DNA content distributions by flow cytometry (***Figure 5—figure supplement 1B***). Using low-coverage scDNAseq data, we characterized the karyotypes of 36 DMSO- and 134 paclitaxel-treated cells. As expected, virtually all cells had extensive aneuploidy after paclitaxel, in contrast with low variance in the control (***Figure 5B***). Additionally, the mean of the resultant aneuploid karyotypes for each chromosome still resembled those of bulk-sequenced cells, highlighting that bulk-sequencing is an ensemble average, and does not detect variation in population aneuploidy, particularly with balanced mis-segregation events (***Figure 5B***, single-cell mean and bulk). In quantifying the absolute deviation from the modal control karyotype in each cell, and assuming a single mitosis, cells exposed to 20 nM paclitaxel mis-segregate 18.5 ± 0.5—a $P_{misseg}$ of ~0.42 considering the control's sub-diploid modal karyotype (***Figure 5C***). The majority of these appeared to be whole-chromosome mis-segregations (***Figure 5—figure supplement 2***).

In this instance, we were able to estimate mis-segregation rate by calculating absolute deviation from the modal karyotype after a single aberrant cell division. However, such an analysis would not be possible for long-term experiments, or real tumors, where new aneuploid cells may be subject to selection. Accordingly, we sought to infer the parameters of this experiment—the mis-segregation rate of 18.5 chromosomes per division and low selection—using only measures of aneuploidy, variance, and phylogenetic topology. To display this, we used dimensionality reduction to ensure that observed measures from the paclitaxel-treated Cal51 population fell within the space of those observed from

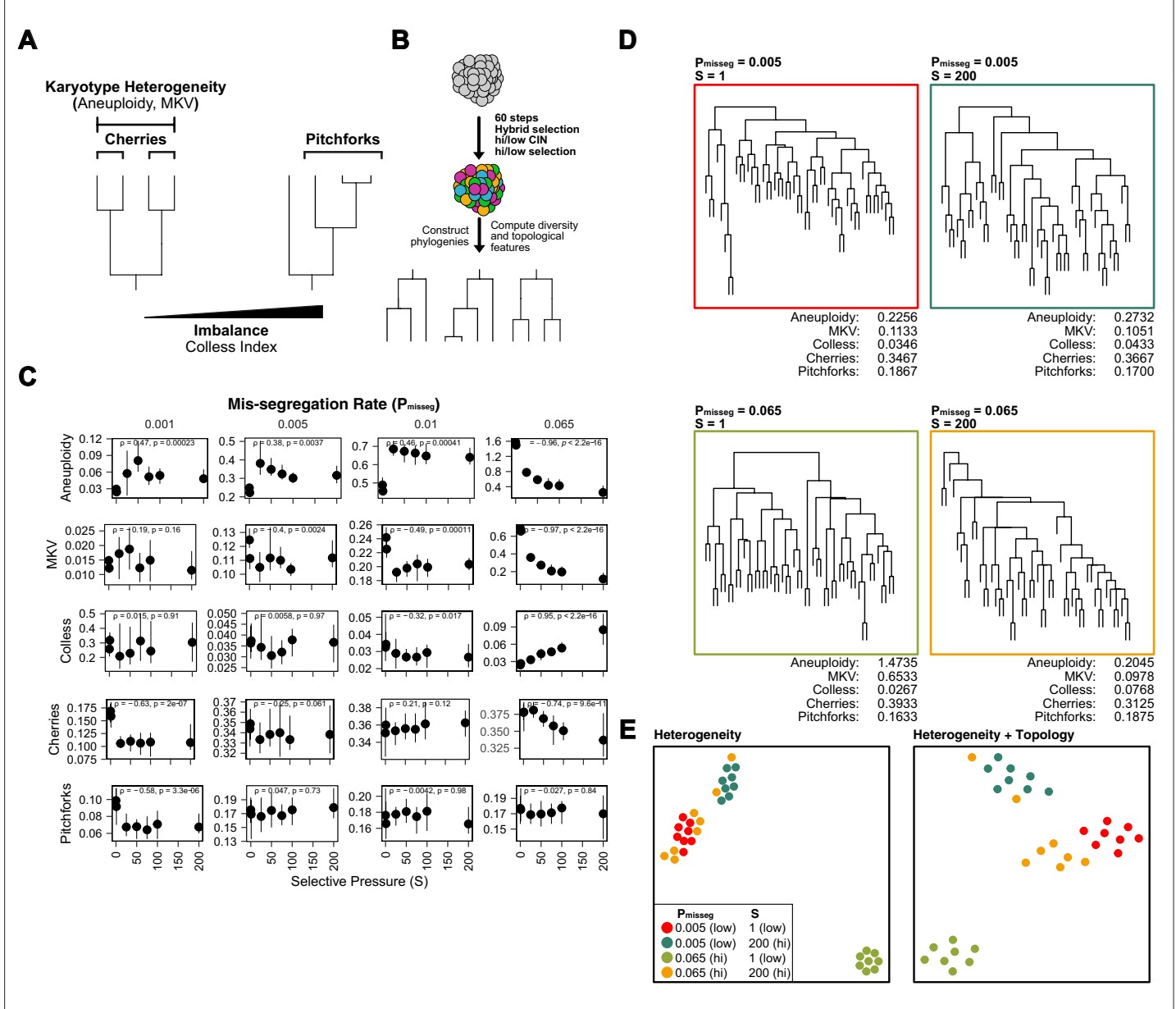

**Figure 4.** Topological features of simulated phylogenies delineate CIN rate and karyotype selection. (**A**) Quantifiable features of karyotypically diverse populations. Heterogeneity between and within karyotypes is described by MKV and aneuploidy (inter- and intra-karyotype variance, see Materials and methods). We also quantify discrete topological features of phylogenetic trees, such as cherries (tip pairs) and pitchforks (3-tip groups), and a whole-tree measure of imbalance (or asymmetry), the Colless index. (**B**) Scheme to test how CIN and selection influence the phylogenetic topology of simulated populations. (**C**) Computed heterogeneity (aneuploidy and MKV) and topology (Colless index, cherries, pitchforks) summary statistics under varying $P_{misseg}$ and $S$ values. MKV is normalized to the average ploidy of the population. Topological measures are normalized to population size. Spearman rank correlation coefficients (r) and p-values are displayed (n = 8 simulations). (**D**) Representative phylogenies for each hi/low CIN, hi/low selection parameter combination and their computed summary statistics. Each phylogeny represents n = 50 out of 300 cells for each simulation. (**E**) Dimensionality reduction of all simulations for each hi/low CIN, hi/low selection parameter combination using measures of karyotype heterogeneity only (left; MKV and aneuploidy) or measures of karyotype heterogeneity and phylogenetic topology (right; MKV, aneuploidy, Colless index, cherries, and pitchforks).

simulated populations over 2 steps under the Hybrid model. The experimental data mapped to those from simulations using high mis-segregation rates and relatively low selection (red point, *Figure 5D*). However, this comparison does not provide a quantitative measure of CIN. Instead, parameter inference via approximate Bayesian computation (ABC) is well suited for this purpose.

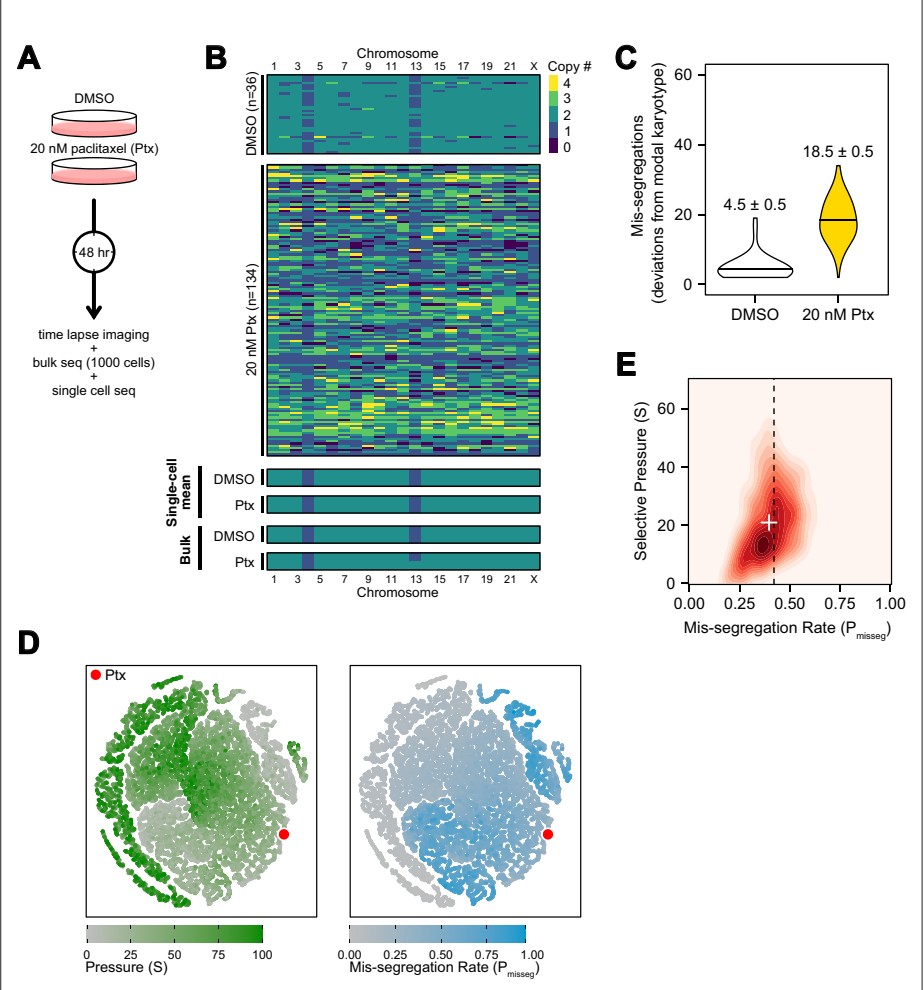

**Figure 5.** Experimental chromosome mis-segregation measured by Bayesian inference experimental scheme. (**A**) Cal51 cells were treated with either DMSO or 20 nM paclitaxel for 48 hr prior to further analysis by time lapse imaging, bulk DNA sequencing, and scDNAseq. (**B**) Heatmaps showing copy number profiles derived from scDNAseq data, single-cell copy number averages, and bulk DNA sequencing. (**C**) Observed mis-segregations calculated as the absolute sum of deviations from the observed modal karyotype of the control. (**D**) Dimensionality reduction analysis of population summary statistics (aneuploidy, MKV, Colless index, cherries) from the first three time steps of all simulations performed under the Hybrid model. (**E**) 2D density plot showing joint posterior distributions from ABC analysis using population summary statistics computed from the paclitaxel-treated cells using the following priors and parameters: Growth Model = 'exponential pseudo-Moran', Selection Model = 'Hybrid, initial ploidy = 2, 2 time steps, S ∈[0, 2... 100], $P_{misseg}$∈[0, 0.005... 1.00] and a tolerance threshold of 0.05 to reject dissimilar simulation results. (see Materials and Methods). Vertical dashed line represents the experimentally observed mis-segregation rate. White + represents the mean of inferred values.

The online version of this article includes the following figure supplement(s) for figure 5:

**Figure supplement 1.** Induction of extensive chromosome mis-segregation via paclitaxel.

**Figure supplement 2.** Copy number profiles of DMSO- and paclitaxel-treated Cal51 cells.

**Figure supplement 3.** Summary statistic optimization for ABC.

**Figure supplement 4.** Nullisomy and posterior predictive checks of summary statistics from paclitaxel-treated Cal51 cells.

**Figure supplement 5.** Minimum sampling of karyotype heterogeneity.

By deriving phylogeny metrics from simulated populations under a wide-range of distributions of evolutionary parameters, ABC identifies evolutionary parameters most consistent with the data—the posterior probability distribution. We used ABC with simulated data to infer the chromosome mis-segregation rate and selective pressure in the paclitaxel-treated cells (*Csilléry et al., 2012*). Importantly, this data has directly observed rates of mis-segregation, which provide a gold standard benchmark to optimize ABC inference.

One key aspect of ABC is the selection of optimal phylogenetic summary statistics. A small number of summary statistics is optimal and larger numbers impair the model (*Csilléry et al., 2012*). To address this, a common approach is to identify a small set of summary statistics that achieve the best inference. Here, we used the experimentally observed mis-segregation rate as a benchmark to optimally select a panel of measures for parameter inference (*Figure 5—figure supplement 3*) and selected the following four metrics to use concurrently in our ABC analysis: mean aneuploidy, MKV, the Colless index (a phylogenetic balance index) and number of cherries (normalized to population size). In doing so, this analysis inferred a chromosome mis-segregation rate of 0.396 ± 0.003 (or 17.4 ± 0.1 chromosomes; mean ± SE), which compares favorably with the experimentally observed rate of 18.5 ± 0.5 (*Figure 5E*; dashed line represents experimental rate, white '+' the inferred rate). The distribution of accepted values for selection was skewed toward lower pressure (21 ± 0.4; mean ± SE), meaning that karyotype selection had little bearing on the result at this time point, consistent with the absence of selection in a 48-hr experiment.

Interestingly, the incidence of nullisomy in the simulated population was higher than in the paclitaxel-treated populations at the observed mis-segregation rate (*Figure 5—figure supplement 4A*). This could be due to spindle pole clustering, a recovery mechanism often seen in paclitaxel-treated cells that causes non-random chromosome mis-segregations. A posterior predictive check of the summary statistics demonstrates how each contributes to the inference of CIN rate (*Figure 5—figure supplement 4B*). In short, this experimental case validated ABC-derived mis-segregation rate as a measure of CIN, with an experimentally determined mis-segregation rate. Importantly, prior estimations of mis-segregation rate selective pressure were not required to develop this quantitative measure of CIN.

Together, these data indicate that combining simulated and observed metrics of population diversity and structure with a Bayesian framework for parameter inference is a flexible method of quantifying the evolutionary forces associated with CIN. Moreover, this method reveals the hitherto unreported potential extent of chromosome mis-segregation induced by a clinically relevant concentration of the successful chemotherapeutic paclitaxel consistent with the measured mis-segregation from non-pharmacologically induced multipolar divisions (*Bollen et al., 2021*).

## Minimum sampling of karyotype heterogeneity

The cost of high-throughput DNA sequencing of single cells is often cited as a limitation to clinical implementation (*Evrony et al., 2021*). In part, the cost can be limited by low-coverage sequencing which is sufficient to estimate the density of reads across the genome. Further, it may be possible to minimize the number of cells that are sampled to get a robust estimate of CIN, though sampling too few cells may result in inaccurate measurements. Accordingly, we determined how sampling impacts measurement of mis-segregation rates using approximate Bayesian computation. We first took five random samples from the population of paclitaxel-treated cells each at various sample sizes (*Figure 5—figure supplement 5A*). We then inferred the mis-segregation rate in each sample and identified the sample size that surpasses an average of 90% accuracy and a low standard error of measurement. We found that even small sample sizes can accurately infer the mis-segregation rate, in this context, with a low standard error (*Figure 5—figure supplement 5B-D*). A sample size of 60 cells produced the most accurate measurement at 99.5% and a standard error of 0.008 ( ± 0.35 chromosomes). We repeated this analysis using simulated data from the Hybrid selection model and a range of mis-segregation rates spanning what is observed in cancer and non-cancer cultures ($P_{misseg} \leq 0.02$; see below). We again found a range of sample sizes whose inferred mis-segregation rates underestimate the known value from those simulations ($n \in [20, 40... 180]$; *Figure 5—figure supplement 5E*,F). Across all mis-segregation rates and selective pressures, random samples of 200 cells had a median percent accuracy of 90% and median standard error of 0.0003 ( ± 0.0138 chromosomes per division). The difference in optimal sample sizes between the paclitaxel-treated population and the simulated population is notable and likely due to the presence of 'clonal' structures in the simulated population.

While the paclitaxel treatment resulted in a uniformly high degree of aneuploidy and little evidence of karyotype selection, the simulated populations after 60 steps (~30 generations) have discrete copy number clusters that may not be captured in each random sample. To verify this, we repeated the analysis using only data from the first time step, prior to the onset of karyotype selection (*Figure 5—figure supplement 5H*). In this case, we found that the sample size needed to achieve a median 90% accuracy over all simulations in this context is 100 cells, at which point the standard error for $P_{misseg}$ is 0.0068 (placing measures within ±0.31 chromosomes per division; *Figure 5—figure supplement 5I, J*). Thus, a larger number of cells is required in the context of long-term karyotype selection than a more acute time scale, such as we see with paclitaxel.

In conclusion, we recommend using 200 cells from a single sampled site which, at biologically relevant time scales and rates of mis-segregation, provides ≥90% accuracy. These data represent, to our knowledge, the first analysis of how sample size for single-cell sequencing affects the accuracy and measurement of chromosome mis-segregation rates.

## Inferring chromosome mis-segregation rates in tumors and organoids

To determine if this framework is clinically applicable, we employed previously published scDNAseq datasets derived from tumor samples and patient-derived organoids (PDO) (*Bolhaqueiro et al., 2019*; *Navin et al., 2011*). Importantly, the data from Bolhaqueiro et al. include sample-matched live cell imaging data in colorectal cancer PDOs, with direct observation of chromosome mis-segregation events to compare with inferred measures. We established our panel of measurements on these populations (*Figure 6A*) and used these to tune the prior distribution of time steps and the rejection threshold for ABC. In sensitivity analysis, 20 steps or greater was sufficient to establish stable estimates of $P_{misseg}$ and selection, S (*Figure 6—figure supplement 1A-B*)—we chose a window of 40–80 steps for further analysis. For rejection thresholds 0.05 and smaller, the inferred mis-segregation rates remained steady (*Figure 6—figure supplement 1C*). With these model parameters chosen, we evaluated the different selection models, and found that the Abundance model resulted in simulated data that best resembled experimental data, for both exponential and constant-population dynamics (*Table 3*). Given that the Abundance model is the most biologically relevant, we will use data simulated under this model in our prior dataset for inference.

Having confirmed the summary statistics from these samples were within the space of the simulation data with our chosen priors (*Figure 6B*), we performed ABC analysis on these datasets to infer rates of chromosome mis-segregation and levels of selection pressure and display the joint posterior distributions as 2D density plots (*Figure 6C and D*; *Figure 6—figure supplements 2 and 3*). *Figure 6C* illustrates the results for two individual colon organoid lines, showing the distribution of parameters used for simulations that gave the most similar results. With ABC, inferred parameters fall within rates of mis-segregation of about 0.001–0.006. Applied to a near-diploid cell, this translates to a range of about 5–38% of cell divisions having one chromosome mis-segregation. Importantly, these inferred rates of chromosome mis-segregation fall within the range of approximated *per chromosome* rates experimentally observed in cancer cell lines and human tumors (*Figure 6E*; *Table 4*, *Table 5*; *Bakhoum et al., 2014*; *Bakhoum et al., 2011*; *Bakhoum et al., 2009*; *Dewhurst et al., 2014*; *Nicholson et al., 2015*; *Orr et al., 2016*; *Thompson and Compton, 2008*; *Worrall et al., 2018*; *Zasadil et al., 2014*). Higher inferred mis-segregation rates tended to coincide with lower inferred selection experienced in these samples (*Figure 6F*). Posterior distributions in these samples were skewed toward high selection (S) indicating the presence stabilizing selection in all cases, where the average of the distributions of some samples were slightly lower or higher (*Figure 6—figure supplement 3*).

To confirm the relevance of the inferred scalar exponent we performed our model selection scheme using only the simulation data with unmodified fitness values (S = 1; *Table 4*). In this case, we found that the inferred mis-segregation rates for most samples fell well below the expected range found in cancer cell lines (*Figure 6E*). Additionally, when we inferred mis-segregation rates and selection in the early timepoint of longitudinally sequenced organoid clones from *Bolhaqueiro et al., 2019*, the composition of the resultant populations simulated using these inferred characteristics better resembled the late-timepoint organoid data than those with unmodified selection values (S = 1; *Figure 6—figure supplements 4 and 5*).

As further validation for mis-segregation rates, we compared these inferred rates from CRC PDOs with those directly measured in live imaging from *Bolhaqueiro et al., 2019*. Although mis-segregation

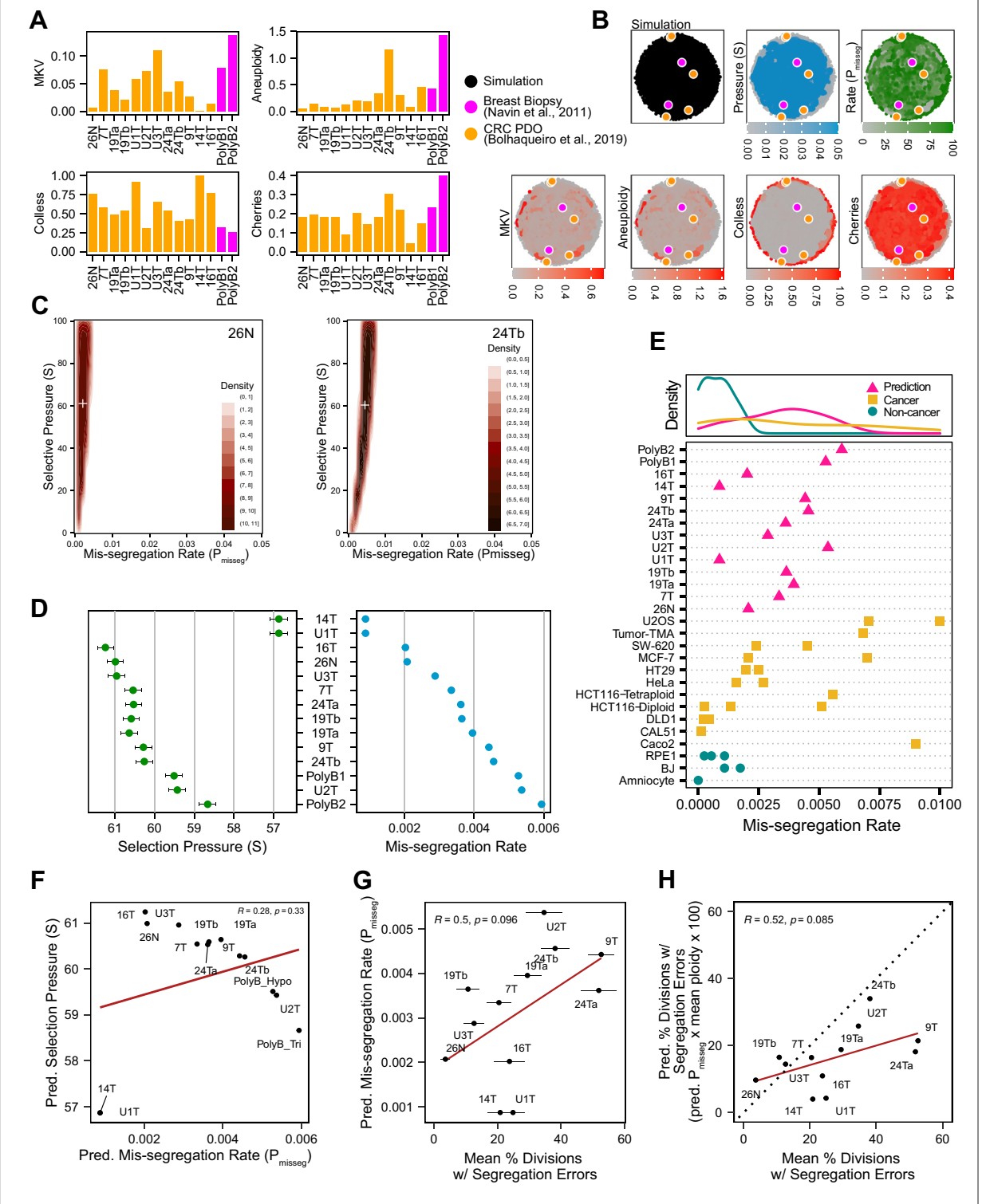

**Figure 6.** Inferring chromosome mis-segregation rates in tumors and organoids *Bolhaqueiro et al., 2019Navin et al., 2011*. (**A**) Computed population summary statistics for colorectal cancer (CRC) patient-derived organoids (PDOs) and breast biopsy scDNAseq datasets from *Bolhaqueiro et al., 2019* (gold) and *Navin et al., 2011* (pink). (**B**) Dimensionality reduction analysis of population summary statistics showing biological observations overlaid on, and found within, the space of simulated observations. Point colors show the simulation parameters and summary statistics for all simulations using the following priors and parameters: Growth Model = 'exponential pseudo-Moran', Selection Model = 'Abundance', initial ploidy = 2, time steps ∈[40, 41… 80], S ∈[0,2… 100], $P_{misseg}$∈[0,0.001… 0.050] and a tolerance threshold of 0.05 to reject dissimilar simulation results. (see Materials and Methods). (**C**) 2D density plots showing joint posterior distributions of $P_{misseg}$ and S values from the approximate Bayesian computation analysis of samples 26 N (left)

*Figure 6 continued on next page*

*Figure 6 continued*

and 24Tb (right) from *Bolhaqueiro et al., 2019*. White + represents the mean of inferred values. (**D**) Inferred selective pressures and mis-segregation rates from each scDNAseq dataset (mean and SEM of accepted values). (**E**) Predicted mis-segregation rates in CRC PDOs and a breast biopsy plotted with approximated mis-segregation rates observed in cancer (blue triangle) and non-cancer (red circle) models (primarily cell lines) from previous studies (*Table 5*; see Materials and methods). The predicted mis-segregation rates in these cancer-derived samples fall within those observed in cancer cell lines and above those of non-cancer cell lines. (**F**) Pearson correlation of predicted mis-segregation rates and predicted selective pressures in CRC PDOs from *Bolhaqueiro et al., 2019*. (**G**) Pearson correlation of predicted mis-segregation rates and the incidence of observed segregation errors in CRC PDOs from *Bolhaqueiro et al., 2019*. Error bars represent SEM values. (**H**) Pearson correlation of observed incidence of segregation errors in CRC PDOs from *Bolhaqueiro et al., 2019* to the ploidy-corrected prediction of the observed incidence of segregation errors. These values assume the involvement of 1 chromosome per observed error and are calculated as the (predicted mis-segregation rate) x (mean number of chromosomes observed per cell) x 100. Dotted line = 1:1 reference.

The online version of this article includes the following figure supplement(s) for figure 6:

**Figure supplement 1.** ABC-inference threshold and step-window analysis.

**Figure supplement 2.** ABC-inferred step count in patient-derived samples.

**Figure supplement 3.** ABC-inferred mis-segregation rates and selective pressures in patient-derived samples.

**Figure supplement 4.** Validation of selection in longitudinally sequenced CRC organoids.

**Figure supplement 5.** Joint posterior distributions from CRC organoids at 3 weeks.

cannot be directly inferred from microscopy, diversity should correlate with the observed rate of mitotic errors. There was a strong correlation but for two outliers—14T and U1T (*Figure 6G*). In fact, when adjusting to the same scale and correcting for cell ploidy, these data follow a strong positive linear trend with a slightly lower slope than a 1:1 correlation, which could reflect an overestimation of mis-segregation rates in the microscopy data (*Figure 6H*). Particularly with lagging chromosomes, despite a chromosome's involvement in an observed segregation defect, it may end up in the correct daughter cell. Overall, these results indicate that the inferred measures using approximate Bayesian computation and scDNAseq account for selection and provide a quantitative measure of CIN.

## Discussion

The clinical assessment of mutations, short indels, and microsatellite instability in human cancer determined by short-read sequencing currently guide clinical care. By contrast, CIN is highly prevalent, yet has remained largely intractable to clinical measures. Single-cell DNA sequencing now promises detailed karyotypic analysis across hundreds of cells, yet selective pressure suppresses the observed karyotype heterogeneity within a tumor. Optimal clinical measurement of CIN may be achieved with scDNAseq, but must additionally account for selective pressure, which reduces karyotype heterogeneity.

Despite the major limitations with current measures of CIN, emerging evidence hints at its utility as a biomarker to predict benefit to cancer therapy. For example, CIN measures appear to predict therapeutic response to paclitaxel (*Janssen et al., 2009*; *Scribano et al., 2021*; *Swanton et al., 2009*). Nevertheless, existing measures of CIN have had significant limitations. FISH and histological analysis of mitotic abnormalities are limited in quantifying specific chromosomes or requiring highly proliferative tumor types, such as lymphomas and leukemia. Gene expression profiles are proposed to correlate with CIN among populations of tumor samples (*Carter et al., 2006*), although they happen to correlate better with tumor proliferation (*Sheltzer, 2013*); in any case, they are correlations across populations of tumors, not suitable as an individualized diagnostic. We conclude that scDNAseq is the most complete and tractable measure of cellular karyotypes, and sampling at least 200 cells, coupled with computational models and ABC, promises to offer the best measure of tumor CIN.

Computational modeling of aneuploidy and CIN has been used to explore evolution in the context of numerical CIN and karyotype selection (*Elizalde et al., 2018*; *Gao et al., 2016*; *Gusev et al., 2001*; *Gusev et al., 2000*; *Laughney et al., 2015*; *Nowak et al., 2002*). Gusev and Nowak lay the foundation for mathematical modeling of CIN. While Gusev focused on the karyotypic outcomes of CIN, Nowak considered the effects of CIN-inducing mutations and the subsequent rate of LOH. Neither considered the individual fitness differences between specific karyotypes (*Gusev et al., 2001*; *Gusev et al., 2000*; *Nowak et al., 2002*). This was improved in *Laughney et al., 2015* and *Elizalde et al., 2018* where the authors leveraged the chromosome scores derived in *Davoli et al., 2013*, which enable the

**Table 3.** Model selection.

| Sample | Growt Model | Selectio Model | PP | BF (Ho Neutral) | Pmisseg | S | Steps |
|---|---|---|---|---|---|---|---|
| 7T | exponential pseudo-Moran | Abundance | 0.621 | Inf | 0.0033 ± 1e-05 | 60.5416 ± 0.2053 | 59.8475 ± 0.0937 |
| 7T | exponential pseudo-Moran | Driver | 0.14 | Inf | 0.001 ± 1e-05 | 49.6557 ± 0.2389 | 58.7002 ± 0.0943 |
| 7T | exponential pseudo-Moran | Hybrid | 0.239 | Inf | 8e-04 ± 1e-05 | 49.3428 ± 0.2377 | 58.5789 ± 0.0935 |
| 7T | exponential pseudo-Moran | Neutral | 0 | NA | 9e-04 ± 5e-05 | 0 ± 0 | 57.7994 ± 0.6728 |
| 7T | constant Wright-Fisher | Abundance | 0.985 | Inf | 0.0062 ± 2e-05 | 69.7026 ± 0.1724 | 59.9318 ± 0.0937 |
| 7T | constant Wright-Fisher | Driver | 0 | NA | 0.0012 ± 1e-05 | 48.2881 ± 0.2384 | 57.5239 ± 0.0933 |
| 7T | constant Wright-Fisher | Hybrid | 0.015 | Inf | 9e-04 ± 1e-05 | 50.7803 ± 0.2359 | 58.2514 ± 0.0941 |
| 7T | constant Wright-Fisher | Neutral | 0 | NA | 9e-04 ± 5e-05 | 0 ± 0 | 58.7803 ± 0.6701 |
| U1T | exponential pseudo-Moran | Abundance | 0.582 | 199 | 9e-04 ± 1e-05 | 56.8672 ± 0.2168 | 59.9906 ± 0.0937 |
| U1T | exponential pseudo-Moran | Driver | 0.113 | 39 | 0.001 ± 1e-05 | 49.6611 ± 0.2389 | 58.6886 ± 0.0944 |
| U1T | exponential pseudo-Moran | Hybrid | 0.156 | 54 | 8e-04 ± 1e-05 | 49.3658 ± 0.2375 | 58.569 ± 0.0935 |
| U1T | exponential pseudo-Moran | Neutral | 0.149 | 1 | 9e-04 ± 5e-05 | 0 ± 0 | 57.7102 ± 0.67 |
| U1T | constant Wright-Fisher | Abundance | 0.654 | 290 | 0.001 ± 1e-05 | 61.4358 ± 0.2029 | 60.0021 ± 0.0937 |
| U1T | constant Wright-Fisher | Driver | 0.115 | 51 | 0.0012 ± 1e-05 | 48.2767 ± 0.2383 | 57.5267 ± 0.0934 |
| U1T | constant Wright-Fisher | Hybrid | 0.115 | 51 | 9e-04 ± 1e-05 | 50.8033 ± 0.2358 | 58.2507 ± 0.0941 |
| U1T | constant Wright-Fisher | Neutral | 0.115 | 1 | 9e-04 ± 5e-05 | 0 ± 0 | 58.7803 ± 0.6701 |
| U2T | exponential pseudo-Moran | Abundance | 0.628 | 251 | 0.0054 ± 1e-05 | 59.4269 ± 0.2108 | 59.8349 ± 0.0935 |
| U2T | exponential pseudo-Moran | Driver | 0.079 | 32 | 0.0027 ± 2e-05 | 50.1513 ± 0.2396 | 57.4538 ± 0.0934 |
| U2T | exponential pseudo-Moran | Hybrid | 0.166 | 66 | 0.0022 ± 2e-05 | 48.7779 ± 0.2413 | 57.7078 ± 0.0934 |
| U2T | exponential pseudo-Moran | Neutral | 0.127 | 1 | 0.0021 ± 7e-05 | 0 ± 0 | 56.8535 ± 0.6619 |
| U2T | constant Wright-Fisher | Abundance | 0.918 | 2817 | 0.0112 ± 3e-05 | 69.7222 ± 0.1703 | 60.0655 ± 0.0934 |
| U2T | constant Wright-Fisher | Driver | 0.001 | 4 | 0.0027 ± 2e-05 | 48.7794 ± 0.2389 | 56.4812 ± 0.0919 |

*Table 3 continued on next page*

Table 3 continued

| Sample | Growt Model | Selectio Model | PP | BF (Ho Neutral) | Pmisseg | S | Steps |
|---|---|---|---|---|---|---|---|
| U2T | constant Wright-Fisher | Hybrid | 0.064 | 196 | 0.0022 ± 1e-05 | 50.9564 ± 0.2379 | 57.1161 ± 0.0925 |
| U2T | constant Wright-Fisher | Neutral | 0.017 | 1 | 0.0022 ± 1e-04 | 0 ± 0 | 57.7898 ± 0.6841 |
| U3T | exponential pseudo-Moran | Abundance | 0.582 | 199 | 0.0029 ± 1e-05 | 60.9557 ± 0.2091 | 59.8273 ± 0.0938 |
| U3T | exponential pseudo-Moran | Driver | 0.113 | 39 | 0.001 ± 1e-05 | 49.6707 ± 0.2389 | 58.6986 ± 0.0944 |
| U3T | exponential pseudo-Moran | Hybrid | 0.156 | 54 | 8e-04 ± 1e-05 | 49.3754 ± 0.2376 | 58.5711 ± 0.0935 |
| U3T | exponential pseudo-Moran | Neutral | 0.149 | 1 | 9e-04 ± 5e-05 | 0 ± 0 | 57.7102 ± 0.67 |
| U3T | constant Wright-Fisher | Abundance | 0.736 | Inf | 0.0052 ± 2e-05 | 69.8357 ± 0.1713 | 59.932 ± 0.0934 |
| U3T | constant Wright-Fisher | Driver | 0.13 | Inf | 0.0012 ± 1e-05 | 48.2864 ± 0.2383 | 57.5385 ± 0.0934 |
| U3T | constant Wright-Fisher | Hybrid | 0.134 | Inf | 9e-04 ± 1e-05 | 50.8219 ± 0.2357 | 58.2482 ± 0.0941 |
| U3T | constant Wright-Fisher | Neutral | 0 | NA | 9e-04 ± 5e-05 | 0 ± 0 | 58.8567 ± 0.6676 |
| 14T | exponential pseudo-Moran | Abundance | 0.582 | 199 | 9e-04 ± 1e-05 | 56.8672 ± 0.2168 | 59.9906 ± 0.0937 |
| 14T | exponential pseudo-Moran | Driver | 0.113 | 39 | 0.001 ± 1e-05 | 49.6614 ± 0.239 | 58.695 ± 0.0944 |
| 14T | exponential pseudo-Moran | Hybrid | 0.156 | 54 | 8e-04 ± 1e-05 | 49.3716 ± 0.2375 | 58.5632 ± 0.0935 |
| 14T | exponential pseudo-Moran | Neutral | 0.149 | 1 | 9e-04 ± 5e-05 | 0 ± 0 | 57.7102 ± 0.67 |
| 14T | constant Wright-Fisher | Abundance | 0.654 | 290 | 0.0011 ± 1e-05 | 62.8579 ± 0.2075 | 60.0029 ± 0.0936 |
| 14T | constant Wright-Fisher | Driver | 0.115 | 51 | 0.0012 ± 1e-05 | 48.2967 ± 0.2383 | 57.5295 ± 0.0934 |
| 14T | constant Wright-Fisher | Hybrid | 0.115 | 51 | 9e-04 ± 1e-05 | 50.8274 ± 0.2357 | 58.2478 ± 0.0941 |
| 14T | constant Wright-Fisher | Neutral | 0.115 | 1 | 9e-04 ± 5e-05 | 0 ± 0 | 58.8567 ± 0.6676 |
| 16T | exponential pseudo-Moran | Abundance | 0.582 | 199 | 0.002 ± 1e-05 | 61.2401 ± 0.2028 | 59.9109 ± 0.0935 |
| 16T | exponential pseudo-Moran | Driver | 0.113 | 39 | 0.001 ± 1e-05 | 49.6539 ± 0.2389 | 58.7006 ± 0.0943 |
| 16T | exponential pseudo-Moran | Hybrid | 0.156 | 54 | 8e-04 ± 1e-05 | 49.3611 ± 0.2376 | 58.574 ± 0.0935 |
| 16T | exponential pseudo-Moran | Neutral | 0.149 | 1 | 9e-04 ± 5e-05 | 0 ± 0 | 57.7994 ± 0.6728 |

*Table 3 continued*

| Sample | Growt Model | Selectio Model | PP | BF (Ho Neutral) | Pmisseg | S | Steps |
|---|---|---|---|---|---|---|---|
| 16T | constant Wright-Fisher | Abundance | 0.654 | 290 | 0.0038 ± 1e-05 | 69.8456 ± 0.1701 | 59.9523 ± 0.0936 |
| 16T | constant Wright-Fisher | Driver | 0.115 | 51 | 0.0012 ± 1e-05 | 48.261 ± 0.2384 | 57.5233 ± 0.0933 |
| 16T | constant Wright-Fisher | Hybrid | 0.115 | 51 | 9e-04 ± 1e-05 | 50.7713 ± 0.2359 | 58.2554 ± 0.0941 |
| 16T | constant Wright-Fisher | Neutral | 0.115 | 1 | 9e-04 ± 5e-05 | 0 ± 0 | 58.7803 ± 0.6701 |
| 19Ta | exponential pseudo-Moran | Abundance | 0.711 | 313 | 0.004 ± 1e-05 | 60.6391 ± 0.2074 | 59.7801 ± 0.0934 |
| 19Ta | exponential pseudo-Moran | Driver | 0.038 | 17 | 0.0028 ± 2e-05 | 50.2185 ± 0.2399 | 57.3764 ± 0.0934 |
| 19Ta | exponential pseudo-Moran | Hybrid | 0.135 | 59 | 0.0022 ± 3e-05 | 48.3823 ± 0.242 | 57.5368 ± 0.0935 |
| 19Ta | exponential pseudo-Moran | Neutral | 0.116 | 1 | 0.0022 ± 9e-05 | 0 ± 0 | 56.5955 ± 0.6549 |
| 19Ta | constant Wright-Fisher | Abundance | 0.97 | 11760 | 0.0075 ± 2e-05 | 69.3863 ± 0.1735 | 59.956 ± 0.0938 |
| 19Ta | constant Wright-Fisher | Driver | 0 | 0 | 0.0028 ± 2e-05 | 48.8413 ± 0.2392 | 56.4529 ± 0.0917 |
| 19Ta | constant Wright-Fisher | Hybrid | 0.026 | 315 | 0.0023 ± 1e-05 | 50.8588 ± 0.2383 | 57.1031 ± 0.0925 |
| 19Ta | constant Wright-Fisher | Neutral | 0.004 | 1 | 0.0023 ± 1e-04 | 0 ± 0 | 57.9522 ± 0.6869 |
| 19Tb | exponential pseudo-Moran | Abundance | 0.727 | 320 | 0.0036 ± 1e-05 | 60.5885 ± 0.2085 | 59.829 ± 0.0938 |
| 19Tb | exponential pseudo-Moran | Driver | 0.03 | 13 | 0.001 ± 1e-05 | 49.6622 ± 0.2389 | 58.6929 ± 0.0944 |
| 19Tb | exponential pseudo-Moran | Hybrid | 0.127 | 56 | 8e-04 ± 1e-05 | 48.5237 ± 0.2322 | 58.9663 ± 0.0931 |
| 19Tb | exponential pseudo-Moran | Neutral | 0.116 | 1 | 9e-04 ± 5e-05 | 0 ± 0 | 57.7102 ± 0.67 |
| 19Tb | constant Wright-Fisher | Abundance | 0.979 | 47320 | 0.0068 ± 2e-05 | 69.5697 ± 0.173 | 59.9232 ± 0.0935 |
| 19Tb | constant Wright-Fisher | Driver | 0 | 0 | 0.0012 ± 1e-05 | 48.2786 ± 0.2383 | 57.5433 ± 0.0934 |
| 19Tb | constant Wright-Fisher | Hybrid | 0.02 | 982 | 9e-04 ± 1e-05 | 50.8162 ± 0.2357 | 58.2495 ± 0.0941 |
| 19Tb | constant Wright-Fisher | Neutral | 0.001 | 1 | 9e-04 ± 5e-05 | 0 ± 0 | 58.8376 ± 0.669 |
| 24Ta | exponential pseudo-Moran | Abundance | 0.731 | 321 | 0.0036 ± 1e-05 | 60.5303 ± 0.2082 | 59.8208 ± 0.0938 |
| 24Ta | exponential pseudo-Moran | Driver | 0.029 | 13 | 0.001 ± 1e-05 | 49.6703 ± 0.2389 | 58.6938 ± 0.0944 |

*Table 3 continued on next page*

*Table 3 continued*

| Sample | Growt Model | Selectio Model | PP | BF (Ho Neutral) | Pmisseg | S | Steps |
|---|---|---|---|---|---|---|---|
| 24Ta | exponential pseudo-Moran | Hybrid | 0.125 | 55 | 8e-04 ± 1e-05 | 49.3669 ± 0.2376 | 58.5778 ± 0.0935 |
| 24Ta | exponential pseudo-Moran | Neutral | 0.116 | 1 | 9e-04 ± 5e-05 | 0 ± 0 | 57.7102 ± 0.67 |
| 24Ta | constant Wright-Fisher | Abundance | 0.979 | 47346 | 0.0068 ± 2e-05 | 69.6173 ± 0.173 | 59.933 ± 0.0934 |
| 24Ta | constant Wright-Fisher | Driver | 0 | 0 | 0.0012 ± 1e-05 | 48.2789 ± 0.2383 | 57.5377 ± 0.0934 |
| 24Ta | constant Wright-Fisher | Hybrid | 0.02 | 956 | 9e-04 ± 1e-05 | 50.8229 ± 0.2357 | 58.2524 ± 0.0941 |
| 24Ta | constant Wright-Fisher | Neutral | 0.001 | 1 | 9e-04 ± 5e-05 | 0 ± 0 | 58.8567 ± 0.6676 |
| 24Tb | exponential pseudo-Moran | Abundance | 0.68 | 294 | 0.0046 ± 1e-05 | 60.2602 ± 0.2084 | 59.8073 ± 0.0936 |
| 24Tb | exponential pseudo-Moran | Driver | 0.054 | 23 | 0.0031 ± 3e-05 | 50.2981 ± 0.2399 | 57.2927 ± 0.0934 |
| 24Tb | exponential pseudo-Moran | Hybrid | 0.149 | 65 | 0.0025 ± 4e-05 | 48.3833 ± 0.244 | 57.4236 ± 0.0936 |
| 24Tb | exponential pseudo-Moran | Neutral | 0.118 | 1 | 0.0025 ± 0.00013 | 0 ± 0 | 56.7229 ± 0.6579 |
| 24Tb | constant Wright-Fisher | Abundance | 0.954 | 7730 | 0.0215 ± 0.00011 | 33.6703 ± 0.2962 | 59.9064 ± 0.0937 |
| 24Tb | constant Wright-Fisher | Driver | 0 | 2 | 0.003 ± 2e-05 | 48.7528 ± 0.2393 | 56.4175 ± 0.0918 |
| 24Tb | constant Wright-Fisher | Hybrid | 0.039 | 318 | 0.0024 ± 2e-05 | 50.7006 ± 0.2389 | 57.107 ± 0.0925 |
| 24Tb | constant Wright-Fisher | Neutral | 0.006 | 1 | 0.0024 ± 0.00011 | 0 ± 0 | 58.0318 ± 0.6822 |
| 26N | exponential pseudo-Moran | Abundance | 0.582 | 199 | 0.0021 ± 1e-05 | 60.9877 ± 0.2031 | 59.9205 ± 0.0934 |
| 26N | exponential pseudo-Moran | Driver | 0.113 | 39 | 0.001 ± 1e-05 | 49.6389 ± 0.2389 | 58.7018 ± 0.0944 |
| 26N | exponential pseudo-Moran | Hybrid | 0.156 | 54 | 8e-04 ± 1e-05 | 49.3389 ± 0.2377 | 58.5755 ± 0.0935 |
| 26N | exponential pseudo-Moran | Neutral | 0.149 | 1 | 9e-04 ± 5e-05 | 0 ± 0 | 57.7994 ± 0.6728 |
| 26N | constant Wright-Fisher | Abundance | 0.654 | 290 | 0.0039 ± 1e-05 | 69.794 ± 0.1704 | 59.9547 ± 0.0935 |
| 26N | constant Wright-Fisher | Driver | 0.115 | 51 | 0.0012 ± 1e-05 | 48.2849 ± 0.2384 | 57.5175 ± 0.0933 |
| 26N | constant Wright-Fisher | Hybrid | 0.115 | 51 | 9e-04 ± 1e-05 | 50.737 ± 0.2359 | 58.2609 ± 0.0941 |
| 26N | constant Wright-Fisher | Neutral | 0.115 | 1 | 9e-04 ± 5e-05 | 0 ± 0 | 58.7803 ± 0.6701 |

*Table 3 continued*

| Sample | Growt Model | Selectio Model | PP | BF (Ho Neutral) | Pmisseg | S | Steps |
|---|---|---|---|---|---|---|---|
| 9T | exponential pseudo-Moran | Abundance | 0.685 | 299 | 0.0044 ± 1e-05 | 60.2829 ± 0.2086 | 59.7955 ± 0.0936 |
| 9T | exponential pseudo-Moran | Driver | 0.052 | 23 | 0.0029 ± 2e-05 | 50.2323 ± 0.2398 | 57.3657 ± 0.0934 |
| 9T | exponential pseudo-Moran | Hybrid | 0.147 | 64 | 0.0022 ± 3e-05 | 48.3829 ± 0.2422 | 57.5193 ± 0.0936 |
| 9T | exponential pseudo-Moran | Neutral | 0.117 | 1 | 0.0023 ± 9e-05 | 0 ± 0 | 56.6083 ± 0.6581 |
| 9T | constant Wright-Fisher | Abundance | 0.958 | 9299 | 0.0087 ± 2e-05 | 69.6836 ± 0.1724 | 59.926 ± 0.0937 |
| 9T | constant Wright-Fisher | Driver | 0 | 1 | 0.0028 ± 2e-05 | 48.8394 ± 0.2392 | 56.4465 ± 0.0917 |
| 9T | constant Wright-Fisher | Hybrid | 0.037 | 360 | 0.0023 ± 1e-05 | 50.8477 ± 0.2384 | 57.0952 ± 0.0925 |
| 9T | constant Wright-Fisher | Neutral | 0.005 | 1 | 0.0023 ± 1e-04 | 0 ± 0 | 57.9427 ± 0.687 |
| PolyB1 | exponential pseudo-Moran | Abundance | 0.635 | 261 | 0.0053 ± 1e-05 | 59.5088 ± 0.2104 | 59.8379 ± 0.0935 |
| PolyB1 | exponential pseudo-Moran | Driver | 0.076 | 31 | 0.0028 ± 2e-05 | 50.2364 ± 0.2398 | 57.4025 ± 0.0934 |
| PolyB1 | exponential pseudo-Moran | Hybrid | 0.164 | 67 | 0.0022 ± 3e-05 | 48.6949 ± 0.2419 | 57.6322 ± 0.0934 |
| PolyB1 | exponential pseudo-Moran | Neutral | 0.124 | 1 | 0.0022 ± 9e-05 | 0 ± 0 | 56.5955 ± 0.6549 |
| PolyB1 | constant Wright-Fisher | Abundance | 0.925 | 3482 | 0.0111 ± 3e-05 | 70.2557 ± 0.169 | 60.042 ± 0.0936 |
| PolyB1 | constant Wright-Fisher | Driver | 0.001 | 4 | 0.0028 ± 2e-05 | 48.8194 ± 0.2391 | 56.4451 ± 0.0917 |
| PolyB1 | constant Wright-Fisher | Hybrid | 0.061 | 228 | 0.0023 ± 1e-05 | 50.895 ± 0.2381 | 57.1073 ± 0.0925 |
| PolyB1 | constant Wright-Fisher | Neutral | 0.014 | 1 | 0.0023 ± 1e-04 | 0 ± 0 | 57.9809 ± 0.6861 |
| PolyB2 | exponential pseudo-Moran | Abundance | 0.603 | 218 | 0.0059 ± 1e-05 | 58.6612 ± 0.212 | 59.7835 ± 0.0937 |
| PolyB2 | exponential pseudo-Moran | Driver | 0.086 | 31 | 0.0038 ± 4e-05 | 50.2948 ± 0.2394 | 57.0217 ± 0.093 |
| PolyB2 | exponential pseudo-Moran | Hybrid | 0.17 | 61 | 0.004 ± 7e-05 | 48.9466 ± 0.2472 | 57.28 ± 0.0942 |
| PolyB2 | exponential pseudo-Moran | Neutral | 0.141 | 1 | 0.0033 ± 0.00022 | 0 ± 0 | 56.5732 ± 0.6597 |
| PolyB2 | constant Wright-Fisher | Abundance | 0.893 | 1277 | 0.0301 ± 1e-04 | 3.0543 ± 0.0165 | 59.9142 ± 0.0936 |
| PolyB2 | constant Wright-Fisher | Driver | 0.003 | 4 | 0.0034 ± 3e-05 | 48.7328 ± 0.2396 | 56.3664 ± 0.0917 |

*Table 3 continued on next page*

*Table 3 continued*

| Sample | Growt Model | Selectio Model | PP | BF (Ho Neutral) | Pmisseg | S | Steps |
|--------|-------------|----------------|-----|-----------------|---------|---|-------|
| PolyB2 | constant Wright-Fisher | Hybrid | 0.069 | 98 | 0.0027 ± 2e-05 | 50.3534 ± 0.2405 | 57.1445 ± 0.0928 |
| PolyB2 | constant Wright-Fisher | Neutral | 0.036 | 1 | 0.0026 ± 0.00014 | 0 ± 0 | 58.1592 ± 0.6741 |

**Table 4.** Model selection with selective pressure constrained to S = 1.

| Sample | Growth Model | Selection Model | PP | BF (Ho Neutral) | Pmisseg | S | Steps |
|---|---|---|---|---|---|---|---|
| 7T | exponential pseudo-Moran | Abundance | 0.274 | 1 | 9e-04 ± 5e-05 | 1 ± 0 | 58.2452 ± 0.6646 |
| 7T | exponential pseudo-Moran | Driver | 0.238 | 1 | 9e-04 ± 5e-05 | 1 ± 0 | 58.4745 ± 0.6725 |
| 7T | exponential pseudo-Moran | Hybrid | 0.26 | 1 | 9e-04 ± 5e-05 | 1 ± 0 | 58.586 ± 0.6668 |
| 7T | exponential pseudo-Moran | Neutral | 0.228 | 1 | 9e-04 ± 6e-05 | 1 ± 0 | 58.5446 ± 0.6791 |
| 7T | constant Wright-Fisher | Abundance | 0.259 | 1 | 9e-04 ± 6e-05 | 1 ± 0 | 58.8089 ± 0.6627 |
| 7T | constant Wright-Fisher | Driver | 0.24 | 1 | 9e-04 ± 6e-05 | 1 ± 0 | 58.1783 ± 0.6771 |
| 7T | constant Wright-Fisher | Hybrid | 0.257 | 1 | 9e-04 ± 5e-05 | 1 ± 0 | 59.0924 ± 0.6742 |
| 7T | constant Wright-Fisher | Neutral | 0.245 | 1 | 9e-04 ± 7e-05 | 1 ± 0 | 58.7516 ± 0.6787 |
| U1T | exponential pseudo-Moran | Abundance | 0.275 | 1 | 9e-04 ± 5e-05 | 1 ± 0 | 58.2452 ± 0.6646 |
| U1T | exponential pseudo-Moran | Driver | 0.239 | 1 | 9e-04 ± 5e-05 | 1 ± 0 | 58.4745 ± 0.6725 |
| U1T | exponential pseudo-Moran | Hybrid | 0.258 | 1 | 9e-04 ± 5e-05 | 1 ± 0 | 58.586 ± 0.6668 |
| U1T | exponential pseudo-Moran | Neutral | 0.228 | 1 | 9e-04 ± 6e-05 | 1 ± 0 | 58.5446 ± 0.6791 |
| U1T | constant Wright-Fisher | Abundance | 0.259 | 1 | 9e-04 ± 6e-05 | 1 ± 0 | 58.8089 ± 0.6627 |
| U1T | constant Wright-Fisher | Driver | 0.24 | 1 | 9e-04 ± 6e-05 | 1 ± 0 | 58.1783 ± 0.6771 |
| U1T | constant Wright-Fisher | Hybrid | 0.257 | 1 | 9e-04 ± 5e-05 | 1 ± 0 | 59.1592 ± 0.6715 |
| U1T | constant Wright-Fisher | Neutral | 0.245 | 1 | 9e-04 ± 7e-05 | 1 ± 0 | 58.7516 ± 0.6787 |
| U2T | exponential pseudo-Moran | Abundance | 0.276 | 1 | 0.0021 ± 8e-05 | 1 ± 0 | 57.3057 ± 0.653 |
| U2T | exponential pseudo-Moran | Driver | 0.235 | 1 | 0.0024 ± 0.00011 | 1 ± 0 | 57.7452 ± 0.6634 |
| U2T | exponential pseudo-Moran | Hybrid | 0.264 | 1 | 0.0021 ± 7e-05 | 1 ± 0 | 58.1274 ± 0.654 |
| U2T | exponential pseudo-Moran | Neutral | 0.225 | 1 | 0.0024 ± 0.00011 | 1 ± 0 | 57.8758 ± 0.6772 |
| U2T | constant Wright-Fisher | Abundance | 0.269 | 1 | 0.0023 ± 1e-04 | 1 ± 0 | 58.3439 ± 0.6532 |
| U2T | constant Wright-Fisher | Driver | 0.233 | 1 | 0.0023 ± 9e-05 | 1 ± 0 | 57.4777 ± 0.693 |

*Table 4 continued on next page*

*Table 4 continued*

| Sample | Growth Model | Selection Model | PP | BF (Ho Neutral) | Pmisseg | S | Steps |
|---|---|---|---|---|---|---|---|
| U2T | constant Wright-Fisher | Hybrid | 0.263 | 1 | 0.0023 ± 1e-04 | 1 ± 0 | 57.8662 ± 0.6683 |
| U2T | constant Wright-Fisher | Neutral | 0.236 | 1 | 0.0025 ± 0.00012 | 1 ± 0 | 57.1433 ± 0.6655 |
| U3T | exponential pseudo-Moran | Abundance | 0.275 | 1 | 9e-04 ± 5e-05 | 1 ± 0 | 58.1624 ± 0.6643 |
| U3T | exponential pseudo-Moran | Driver | 0.239 | 1 | 9e-04 ± 5e-05 | 1 ± 0 | 58.4554 ± 0.6736 |
| U3T | exponential pseudo-Moran | Hybrid | 0.258 | 1 | 9e-04 ± 5e-05 | 1 ± 0 | 58.586 ± 0.6668 |
| U3T | exponential pseudo-Moran | Neutral | 0.228 | 1 | 9e-04 ± 6e-05 | 1 ± 0 | 58.6178 ± 0.6777 |
| U3T | constant Wright-Fisher | Abundance | 0.259 | 1 | 9e-04 ± 6e-05 | 1 ± 0 | 58.7611 ± 0.6614 |
| U3T | constant Wright-Fisher | Driver | 0.24 | 1 | 9e-04 ± 6e-05 | 1 ± 0 | 58.1783 ± 0.6771 |
| U3T | constant Wright-Fisher | Hybrid | 0.257 | 1 | 9e-04 ± 5e-05 | 1 ± 0 | 59.0955 ± 0.674 |
| U3T | constant Wright-Fisher | Neutral | 0.245 | 1 | 9e-04 ± 7e-05 | 1 ± 0 | 58.7516 ± 0.6787 |
| 14T | exponential pseudo-Moran | Abundance | 0.275 | 1 | 9e-04 ± 5e-05 | 1 ± 0 | 58.1624 ± 0.6643 |
| 14T | exponential pseudo-Moran | Driver | 0.239 | 1 | 9e-04 ± 5e-05 | 1 ± 0 | 58.4554 ± 0.6736 |
| 14T | exponential pseudo-Moran | Hybrid | 0.258 | 1 | 9e-04 ± 5e-05 | 1 ± 0 | 58.586 ± 0.6668 |
| 14T | exponential pseudo-Moran | Neutral | 0.228 | 1 | 9e-04 ± 6e-05 | 1 ± 0 | 58.5446 ± 0.6791 |
| 14T | constant Wright-Fisher | Abundance | 0.259 | 1 | 9e-04 ± 6e-05 | 1 ± 0 | 58.8089 ± 0.6627 |
| 14T | constant Wright-Fisher | Driver | 0.24 | 1 | 9e-04 ± 6e-05 | 1 ± 0 | 58.1783 ± 0.6771 |
| 14T | constant Wright-Fisher | Hybrid | 0.257 | 1 | 9e-04 ± 5e-05 | 1 ± 0 | 59.0924 ± 0.6739 |
| 14T | constant Wright-Fisher | Neutral | 0.245 | 1 | 9e-04 ± 7e-05 | 1 ± 0 | 58.7516 ± 0.6787 |
| 16T | exponential pseudo-Moran | Abundance | 0.274 | 1 | 9e-04 ± 5e-05 | 1 ± 0 | 58.2452 ± 0.6646 |
| 16T | exponential pseudo-Moran | Driver | 0.238 | 1 | 9e-04 ± 5e-05 | 1 ± 0 | 58.4745 ± 0.6725 |
| 16T | exponential pseudo-Moran | Hybrid | 0.26 | 1 | 9e-04 ± 5e-05 | 1 ± 0 | 58.586 ± 0.6668 |
| 16T | exponential pseudo-Moran | Neutral | 0.228 | 1 | 0.001 ± 6e-05 | 1 ± 0 | 58.6274 ± 0.6789 |

Table 4 continued

| Sample | Growth Model | Selection Model | PP | BF (Ho Neutral) | Pmisseg | S | Steps |
|---|---|---|---|---|---|---|---|
| 16T | constant Wright-Fisher | Abundance | 0.259 | 1 | 9e-04 ± 6e-05 | 1 ± 0 | 58.8089 ± 0.6627 |
| 16T | constant Wright-Fisher | Driver | 0.24 | 1 | 9e-04 ± 6e-05 | 1 ± 0 | 58.1783 ± 0.6771 |
| 16T | constant Wright-Fisher | Hybrid | 0.257 | 1 | 9e-04 ± 5e-05 | 1 ± 0 | 59.1051 ± 0.6742 |
| 16T | constant Wright-Fisher | Neutral | 0.245 | 1 | 9e-04 ± 7e-05 | 1 ± 0 | 58.7516 ± 0.6787 |
| 19Ta | exponential pseudo-Moran | Abundance | 0.273 | 1 | 0.0021 ± 8e-05 | 1 ± 0 | 57.4045 ± 0.6565 |
| 19Ta | exponential pseudo-Moran | Driver | 0.243 | 1 | 0.0024 ± 0.00011 | 1 ± 0 | 57.8025 ± 0.663 |
| 19Ta | exponential pseudo-Moran | Hybrid | 0.261 | 1 | 0.0022 ± 8e-05 | 1 ± 0 | 57.9108 ± 0.65 |
| 19Ta | exponential pseudo-Moran | Neutral | 0.222 | 1 | 0.0025 ± 0.00012 | 1 ± 0 | 57.9331 ± 0.6777 |
| 19Ta | constant Wright-Fisher | Abundance | 0.27 | 1 | 0.0024 ± 0.00011 | 1 ± 0 | 58.2866 ± 0.6566 |
| 19Ta | constant Wright-Fisher | Driver | 0.233 | 1 | 0.0023 ± 1e-04 | 1 ± 0 | 57.8185 ± 0.6927 |
| 19Ta | constant Wright-Fisher | Hybrid | 0.261 | 1 | 0.0023 ± 1e-04 | 1 ± 0 | 58.0478 ± 0.6705 |
| 19Ta | constant Wright-Fisher | Neutral | 0.237 | 1 | 0.0025 ± 0.00012 | 1 ± 0 | 57.2261 ± 0.6669 |
| 19Tb | exponential pseudo-Moran | Abundance | 0.275 | 1 | 9e-04 ± 5e-05 | 1 ± 0 | 58.1624 ± 0.6643 |
| 19Tb | exponential pseudo-Moran | Driver | 0.239 | 1 | 9e-04 ± 5e-05 | 1 ± 0 | 58.4554 ± 0.6736 |
| 19Tb | exponential pseudo-Moran | Hybrid | 0.258 | 1 | 9e-04 ± 5e-05 | 1 ± 0 | 58.586 ± 0.6668 |
| 19Tb | exponential pseudo-Moran | Neutral | 0.228 | 1 | 9e-04 ± 6e-05 | 1 ± 0 | 58.5796 ± 0.6796 |
| 19Tb | constant Wright-Fisher | Abundance | 0.259 | 1 | 9e-04 ± 6e-05 | 1 ± 0 | 58.7611 ± 0.6614 |
| 19Tb | constant Wright-Fisher | Driver | 0.24 | 1 | 9e-04 ± 6e-05 | 1 ± 0 | 58.1178 ± 0.679 |
| 19Tb | constant Wright-Fisher | Hybrid | 0.257 | 1 | 9e-04 ± 5e-05 | 1 ± 0 | 59.1592 ± 0.6715 |
| 19Tb | constant Wright-Fisher | Neutral | 0.245 | 1 | 9e-04 ± 7e-05 | 1 ± 0 | 58.7516 ± 0.6787 |
| 24Ta | exponential pseudo-Moran | Abundance | 0.275 | 1 | 9e-04 ± 5e-05 | 1 ± 0 | 58.1624 ± 0.6643 |
| 24Ta | exponential pseudo-Moran | Driver | 0.239 | 1 | 9e-04 ± 5e-05 | 1 ± 0 | 58.4554 ± 0.6736 |

*Table 4 continued*

| Sample | Growth Model | Selection Model | PP | BF (Ho Neutral) | Pmisseg | S | Steps |
|---|---|---|---|---|---|---|---|
| 24Ta | exponential pseudo-Moran | Hybrid | 0.258 | 1 | 9e-04 ± 5e-05 | 1 ± 0 | 58.586 ± 0.6668 |
| 24Ta | exponential pseudo-Moran | Neutral | 0.228 | 1 | 9e-04 ± 6e-05 | 1 ± 0 | 58.6656 ± 0.6783 |
| 24Ta | constant Wright-Fisher | Abundance | 0.259 | 1 | 9e-04 ± 6e-05 | 1 ± 0 | 58.7611 ± 0.6614 |
| 24Ta | constant Wright-Fisher | Driver | 0.24 | 1 | 9e-04 ± 6e-05 | 1 ± 0 | 58.1783 ± 0.6771 |
| 24Ta | constant Wright-Fisher | Hybrid | 0.257 | 1 | 9e-04 ± 5e-05 | 1 ± 0 | 59.1592 ± 0.6715 |
| 24Ta | constant Wright-Fisher | Neutral | 0.245 | 1 | 9e-04 ± 7e-05 | 1 ± 0 | 58.7516 ± 0.6787 |
| 24Tb | exponential pseudo-Moran | Abundance | 0.273 | 1 | 0.0023 ± 0.00011 | 1 ± 0 | 57.0446 ± 0.6526 |
| 24Tb | exponential pseudo-Moran | Driver | 0.242 | 1 | 0.0025 ± 0.00012 | 1 ± 0 | 57.551 ± 0.6661 |
| 24Tb | exponential pseudo-Moran | Hybrid | 0.264 | 1 | 0.0022 ± 9e-05 | 1 ± 0 | 57.9108 ± 0.6512 |
| 24Tb | exponential pseudo-Moran | Neutral | 0.222 | 1 | 0.0026 ± 0.00013 | 1 ± 0 | 57.7516 ± 0.6758 |
| 24Tb | constant Wright-Fisher | Abundance | 0.267 | 1 | 0.0024 ± 0.00013 | 1 ± 0 | 58.379 ± 0.6601 |
| 24Tb | constant Wright-Fisher | Driver | 0.237 | 1 | 0.0024 ± 1e-04 | 1 ± 0 | 57.7357 ± 0.6922 |
| 24Tb | constant Wright-Fisher | Hybrid | 0.257 | 1 | 0.0023 ± 1e-04 | 1 ± 0 | 57.9045 ± 0.6718 |
| 24Tb | constant Wright-Fisher | Neutral | 0.239 | 1 | 0.0025 ± 0.00012 | 1 ± 0 | 57.2643 ± 0.6726 |
| 26N | exponential pseudo-Moran | Abundance | 0.274 | 1 | 9e-04 ± 5e-05 | 1 ± 0 | 58.2452 ± 0.6646 |
| 26N | exponential pseudo-Moran | Driver | 0.239 | 1 | 9e-04 ± 5e-05 | 1 ± 0 | 58.4045 ± 0.6706 |
| 26N | exponential pseudo-Moran | Hybrid | 0.26 | 1 | 9e-04 ± 5e-05 | 1 ± 0 | 58.586 ± 0.6668 |
| 26N | exponential pseudo-Moran | Neutral | 0.227 | 1 | 0.001 ± 7e-05 | 1 ± 0 | 58.6815 ± 0.6776 |
| 26N | constant Wright-Fisher | Abundance | 0.259 | 1 | 9e-04 ± 6e-05 | 1 ± 0 | 58.8089 ± 0.6627 |
| 26N | constant Wright-Fisher | Driver | 0.239 | 1 | 9e-04 ± 6e-05 | 1 ± 0 | 58.1783 ± 0.6771 |
| 26N | constant Wright-Fisher | Hybrid | 0.257 | 1 | 9e-04 ± 5e-05 | 1 ± 0 | 59.1178 ± 0.6745 |
| 26N | constant Wright-Fisher | Neutral | 0.245 | 1 | 0.001 ± 7e-05 | 1 ± 0 | 58.6879 ± 0.6762 |

*Table 4 continued*

| Sample | Growth Model | Selection Model | PP | BF (Ho Neutral) | Pmisseg | S | Steps |
|---|---|---|---|---|---|---|---|
| 9T | exponential pseudo-Moran | Abundance | 0.274 | 1 | 0.0021 ± 8e-05 | 1 ± 0 | 57.3854 ± 0.6574 |
| 9T | exponential pseudo-Moran | Driver | 0.242 | 1 | 0.0024 ± 0.00011 | 1 ± 0 | 57.8025 ± 0.663 |
| 9T | exponential pseudo-Moran | Hybrid | 0.261 | 1 | 0.0022 ± 8e-05 | 1 ± 0 | 57.9108 ± 0.65 |
| 9T | exponential pseudo-Moran | Neutral | 0.222 | 1 | 0.0025 ± 0.00012 | 1 ± 0 | 57.9522 ± 0.6787 |
| 9T | constant Wright-Fisher | Abundance | 0.269 | 1 | 0.0024 ± 0.00011 | 1 ± 0 | 58.2866 ± 0.6566 |
| 9T | constant Wright-Fisher | Driver | 0.233 | 1 | 0.0023 ± 1e-04 | 1 ± 0 | 57.9076 ± 0.6927 |
| 9T | constant Wright-Fisher | Hybrid | 0.261 | 1 | 0.0023 ± 1e-04 | 1 ± 0 | 58.1115 ± 0.6708 |
| 9T | constant Wright-Fisher | Neutral | 0.236 | 1 | 0.0025 ± 0.00012 | 1 ± 0 | 57.2261 ± 0.6669 |
| PolyB1 | exponential pseudo-Moran | Abundance | 0.274 | 1 | 0.0021 ± 8e-05 | 1 ± 0 | 57.4045 ± 0.6565 |
| PolyB1 | exponential pseudo-Moran | Driver | 0.243 | 1 | 0.0024 ± 0.00011 | 1 ± 0 | 57.7102 ± 0.6622 |
| PolyB1 | exponential pseudo-Moran | Hybrid | 0.261 | 1 | 0.0022 ± 8e-05 | 1 ± 0 | 57.9459 ± 0.6512 |
| PolyB1 | exponential pseudo-Moran | Neutral | 0.222 | 1 | 0.0025 ± 0.00011 | 1 ± 0 | 57.9522 ± 0.6776 |
| PolyB1 | constant Wright-Fisher | Abundance | 0.271 | 1 | 0.0023 ± 0.00011 | 1 ± 0 | 58.2834 ± 0.6575 |
| PolyB1 | constant Wright-Fisher | Driver | 0.231 | 1 | 0.0023 ± 9e-05 | 1 ± 0 | 57.6656 ± 0.6949 |
| PolyB1 | constant Wright-Fisher | Hybrid | 0.261 | 1 | 0.0023 ± 1e-04 | 1 ± 0 | 57.9713 ± 0.6668 |
| PolyB1 | constant Wright-Fisher | Neutral | 0.237 | 1 | 0.0025 ± 0.00012 | 1 ± 0 | 57.207 ± 0.6674 |
| PolyB2 | exponential pseudo-Moran | Abundance | 0.272 | 1 | 0.0027 ± 2e-04 | 1 ± 0 | 56.8471 ± 0.6544 |
| PolyB2 | exponential pseudo-Moran | Driver | 0.245 | 1 | 0.0029 ± 0.00021 | 1 ± 0 | 57.3312 ± 0.6609 |
| PolyB2 | exponential pseudo-Moran | Hybrid | 0.263 | 1 | 0.0024 ± 0.00011 | 1 ± 0 | 57.9204 ± 0.6466 |
| PolyB2 | exponential pseudo-Moran | Neutral | 0.221 | 1 | 0.0029 ± 0.00017 | 1 ± 0 | 57.4236 ± 0.6784 |
| PolyB2 | constant Wright-Fisher | Abundance | 0.268 | 1 | 0.0025 ± 0.00013 | 1 ± 0 | 58.2484 ± 0.6616 |
| PolyB2 | constant Wright-Fisher | Driver | 0.235 | 1 | 0.0026 ± 0.00014 | 1 ± 0 | 57.5796 ± 0.6897 |

*Table 4 continued*

| Sample | Growth Model | Selection Model | PP | BF (Ho Neutral) | Pmisseg | S | Steps |
|--------|--------------|-----------------|-----|-----------------|---------|-----|-------|
| PolyB2 | constant Wright-Fisher | Hybrid | 0.257 | 1 | 0.0026 ± 0.00015 | 1 ± 0 | 58.1115 ± 0.6741 |
| PolyB2 | constant Wright-Fisher | Neutral | 0.24 | 1 | 0.0027 ± 0.00014 | 1 ± 0 | 57.379 ± 0.6701 |

**Table 5.** Approximate reported per chromosome mis-segregation rates.

| 1st Author | DOI | Model | Tumor? | Statistic | Assessment | Approximate observed frequency % | Aprrox modal chromosome # (ATCC) | Approximate mis-segregation rate (per chromosome) |
|---|---|---|---|---|---|---|---|---|
| Bakhoum | https://doi.org/10.1158/1078-0432.CCR-11-2049 | Tumor-TMA | Tumor | Reported | Lagging/Bridging | 31.3 | 46 | 0.00680 |
| Orr | https://doi.org/10.1016/j.celrep.2016.10.030 | U2OS | Tumor | Approx. Mean | Lagging | 32.5 | 46 | 0.00707 |
| Orr | https://doi.org/10.1016/j.celrep.2016.10.030 | HeLa | Tumor | Approx. Mean | Lagging | 22 | 82 | 0.00268 |
| Orr | https://doi.org/10.1016/j.celrep.2016.10.030 | SW-620 | Tumor | Approx. Mean | Lagging | 22.5 | 50 | 0.00450 |
| Orr | https://doi.org/10.1016/j.celrep.2016.10.030 | RPE1 | Non-tumor | Approx. Mean | Lagging | 2.5 | 46 | 0.00054 |
| Orr | https://doi.org/10.1016/j.celrep.2016.10.030 | BJ | Non-tumor | Approx. Mean | Lagging | 8 | 46 | 0.00174 |
| Nicholson | https://doi.org/10.7554/eLife.05068 | Amniocyte | Non-tumor | Approx. Mean | Lagging | 0 | 46 | 0.00000 |
| Nicholson | https://doi.org/10.7554/eLife.05068 | DLD1 | Tumor | Approx. Mean | Lagging | 1 | 46 | 0.00022 |
| Dewhurst | https://doi.org/10.1158/2159-8290.CD-13-0285 | HCT116-Diploid | Tumor | Approx. Mean | Lagging/Bridging | 23 | 45 | 0.00511 |
| Dewhurst | https://doi.org/10.1158/2159-8290.CD-13-0285 | HCT116-Tetraploid | Tumor | Approx. Mean | Lagging/Bridging | 50 | 90 | 0.00556 |
| Bakhoum | https://doi.org/10.1038/ncb1809 | U2OS | Tumor | Reported | Lagging | | 46 | 0.01000 |
| Zasadil | https://doi.org/10.1126/scitranslmed.3007965 | CAL51 | Tumor | Approx. Mean | Lagging | 0.5 | 44 | 0.00011 |
| Thompson | https://doi.org/10.1083/jcb.200712029 | RPE1 | Non-tumor | Approx. Mean | Acute aneuploidy via FISH | | 46 | 0.00025 |
| Thompson | https://doi.org/10.1083/jcb.200712029 | HCT116-Diploid | Tumor | Approx. Mean | Acute aneuploidy via FISH | | 45 | 0.00025 |
| Thompson | https://doi.org/10.1083/jcb.200712029 | HT29 | Tumor | Approx. Mean | Acute aneuploidy via FISH | | 71 | 0.00250 |
| Thompson | https://doi.org/10.1083/jcb.200712029 | Caco2 | Tumor | Approx. Mean | Acute aneuploidy via FISH | | 96 | 0.00900 |
| Thompson | https://doi.org/10.1083/jcb.200712029 | MCF-7 | Tumor | Approx. Mean | Acute aneuploidy via FISH | | 82 | 0.00700 |
| Bakhoum | https://doi.org/10.1016/j.cub.2014.01.019 | HCT116-Diploid | Tumor | Approx. Mean | Lagging | 6 | 45 | 0.00133 |

*Table 5 continued on next page*

*Table 5 continued*

| 1st Author | DOI | Model | Tumor? | Statistic | Assessment | Approximate observed frequency % | Aprrox modal chromosome # (ATCC) | Approximate mis-segregation rate (per chromosome) |
|---|---|---|---|---|---|---|---|---|
| Bakhoum | https://doi.org/10.1016/j.cub.2014.01.019 | DLD1 | Tumor | Approx. Mean | Lagging | 2 | 46 | 0.00043 |
| Bakhoum | https://doi.org/10.1016/j.cub.2014.01.019 | HT29 | Tumor | Approx. Mean | Lagging | 14 | 71 | 0.00197 |
| Bakhoum | https://doi.org/10.1016/j.cub.2014.01.019 | SW-620 | Tumor | Approx. Mean | Lagging | 12 | 50 | 0.00240 |
| Bakhoum | https://doi.org/10.1016/j.cub.2014.01.019 | MCF-7 | Tumor | Approx. Mean | Lagging | 17 | 82 | 0.00207 |
| Bakhoum | https://doi.org/10.1016/j.cub.2014.01.019 | HeLa | Tumor | Approx. Mean | Lagging | 13 | 82 | 0.00159 |
| Worrall | https://doi.org/10.1016/j.celrep.2018.05.047 | BJ | Non-tumor | Approx. Mean | Unspecified Error | 5 | 46 | 0.00109 |
| Worrall | https://doi.org/10.1016/j.celrep.2018.05.047 | RPE1 | Non-tumor | Approx. Mean | Unspecified Error | 5 | 46 | 0.00109 |

inclusion of oncogenes and tumor suppressors in models of CIN as we have done. These studies have provided important insights such as the role of whole-genome doubling as an evolutionary bridge to optimized chromosome stoichiometry. Yet the populations derived in these studies tend to vary to a greater degree than observed with scDNAseq, as they do not model strong selection against aneuploidy. Further, they do not attempt to use their models to measure CIN in biological samples. Here, we build on these models by considering, in addition to the selection on driver genes, the stabilizing selection wrought by chromosomal gene abundance. Further, we consider that the magnitude of selection pressure may not be a constant and implement a modifier to tune selection in our models. Lastly, we use our models as a quantitative measure of CIN that accounts for this selection.

Previous studies using single-cell sequencing identified surprisingly low karyotypic variance in human tumors including breast cancer (*Gao et al., 2016*; *Kim et al., 2018*; *Wang et al., 2014*) and colorectal and ovarian cancer organoids (*Bolhaqueiro et al., 2019*; *Nelson et al., 2020*). It has been difficult to understand these findings in the light of widespread CIN in human cancer (*Sheltzer and Amon, 2011*; *Silk et al., 2013*; *Vasudevan et al., 2020*; *Weaver et al., 2007*; *Weaver and Cleveland, 2009*). The best explanation of this apparent paradox is selection, which moderates karyotypic variance. Accounting for this, we can infer rates of chromosome mis-segregation in tumors or PDOs well within the range of rates observed microscopically in cancer cell lines. Additionally, no previous work, to our knowledge, has estimated the required sample size to infer CIN from scDNAseq data.

As described by others (*Dewhurst et al., 2014*; *López et al., 2020*), and consistent with our findings, early emergence of polyploid cells can markedly reduce apparent selection, leading to an elevated karyotype diversity over time. While we do not explicitly induce chance of whole genome doubling (WGD) events in simulations, populations that begin either diploid or tetraploid converge on near-triploid karyotypes over time, consistent with the notion that WGD can act as an evolutionary bridge to highly aneuploid karyotypes. Notably, our analysis indicates the samples with apparent polyploidy experienced among the lowest levels of karyotype selection.

In some early studies, CIN is considered a binary process—present or absent. We assumed that CIN measures are scalar, not binary, and measure this by rate of chromosome mis-segregation per division. A scalar is appropriate if, for example, there was a consistent probability of chromosome mis-segregation per division. However, we recognize that some mechanisms may not well adhere to this simplified model of CIN. For example, tumors with centrosome amplification may at times undergo bipolar division without mis-segregation, or, at other times, a multipolar division with extensive mis-segregation. Further, it is possible that some mechanisms may have correlated mis-segregations such that a daughter cell that gains one chromosome is more likely to gain other chromosomes, rather than lose them. Another possibility is that CIN could result in the mis-regulation of genes that further modify the rate of CIN. Our model does not yet account for punctuated behavior or changing rates of CIN. Furthermore, while recent studies have reported non-random mis-segregation of chromosomes (*Dumont et al., 2020*; *Worrall et al., 2018*), we did not incorporate these biases into our models as these studies do not reach consensus on which chromosomes are more frequently mis-segregated, which may be model-dependent.

Our approach reconstructs phylogenetic trees via copy number variation (CNV) analysis. This approach may be suboptimal given the selection on aneuploid states, and could be particularly problematic in the setting of convergent evolution. It is possible that this method results in low accuracy of the reconstructed phylogenies. Alternative approaches are possible, but would likely require re-design of the scDNAseq assay to include spiked-in primers that span highly polymorphic regions on each chromosome. If this were done, these sequences could be read in all cells and single-nucleotide polymorphisms could track individual maternal and paternal chromosomes, allowing a means of reconstructing cell phylogeny independent of CNVs. Despite this limitation, our phylogenetic reconstructions did seem to allow inference of CIN measures consistent with directly observed rates of chromosome mis-segregation in our taxol-induced CIN model as well as several independent cancer PDO models and cell lines.

A final limitation of our approach is we used previous estimates of cellular selection in our agent-based model and used these selection models to infer quantitative measures of CIN. While this approach seems to perform well in estimates of mis-segregation rates, we recognize that the selection models do not necessarily represent the real selective pressures on distinct aneuploidies. Future investigations are necessary to measure the selective pressure of distinct aneuploidies—a project that

is now within technological reach. Selective pressures could also be influenced by cell type (*Auslander et al., 2019*; *Dürrbaum et al., 2014*; *Sack et al., 2018*; *Starostik et al., 2020*), tumor cell genetics (*Foijer et al., 2014*; *Grim et al., 2012*; *López-García et al., 2017*; *Simões-Sousa et al., 2018*; *Soto et al., 2017*), and the microenvironment (*Hoevenaar et al., 2020*).

In summation, we developed a theoretical and experimental framework for quantitative measure of chromosomal instability in human cancer. This framework accounts for selective pressure within tumors and employs Approximate Bayesian Computation, a commonly used analysis in evolutionary biology. Additionally, we determined that low-coverage single-cell DNA sequencing of at least 200 cells from a human tumor sample is sufficient to get an accurate ( > 90% accuracy) and reproducible measure of CIN. This work sets the stage for standardized quantitative measures of CIN that promise to clarify the underlying causes, consequences, and clinical utility of this nearly universal form of genomic instability.

# Materials and methods
## Agent-based modeling
Agent-based models were implemented using the agent-based platform, NetLogo 6.0.4 (*Wilensky, 1999*).

### Underlying assumptions for models of CIN and karyotype selection
Chromosome mis-segregation rate is defined as the number of chromosome missegregation events that occur per cellular division.

  Cell division always results in 2 daughter cells.

  Pmisseg,c is assigned uniformly for each cell in a population and for each chromosome.

  Cells die when the copy number of any chromosome is equal to 0 or exceeding 6 unless otherwise noted.

  Steps are based on the rate of division of euploid cells. We assume a probability of division (Pdivision) of 0.5, or half of the population divides every step, for euploid populations. This probabilistic division is to mimic the asynchrony of cellular proliferation and to allow for positive selection, where some cells may divide more rapidly than their euploid ancestors.

  No chromosome is more likely to mis-segregate than any other.

## Chromosome-arm scores
### Gene abundance scores
The R package biomaRt v.2.46.3 was used to pull the chromosome arm location for each gene in Ensembl's 'Human genes' dataset (GRCh38.p13). The number of genes on each chromosome arm were enumerated and Abundance scores were generated by normalizing the number of genes on each chromosome arm by the sum of all enumerated genes across chromosomes. Chromosome arms with no recorded genes were given a score of 0.

### Driver density scores
Arm-level 'TSG-OG-Ess' scores derived in *Davoli et al., 2013* were adapted for our purposes. These values were derived from a pan-cancer analysis (TCGA) of the frequency of mutation of these genes and their location in the genome. These scores correlate with the frequency with which chromosomes are found to be amplified in the genome. We adapted these scores by normalizing the published 'TSG-OG-Ess' score for each chromosome arm by the sum of all Charm scores. Chromosome arms with no published Charm score were given a score of 0. We refer to these as TOE scores for our purposes.

### Hybrid scores
Chromosome arm scores for the Hybrid selection model are the average of the chromosome arm's Gene Abundance and Driver Density scores.

## Implementing karyotype selection
In each model, numerical scores are assigned to each chromosome, the sum of which represents the fitness of the karyotype (*Figure 1B*). At each simulation time step, fitness is re-calculated for each

cell based on its updated karyotype. These fitness values determine if they undergo mitosis in the next round. However, the modality of selection changes how those karyotypes are assessed. Here, we implement four separate karyotype selection models (1) gene abundance, (2) driver density, (3) a hybrid gene abundance and driver density, and (4) neutral selection. The scores that are generated in each produce a fitness value (F) that can then be subjected to pressure (S) as described above.

## Selection on gene abundance

The Gene Abundance selection model relies on the concept of gene dosage stoichiometry where the aneuploid karyotypes are selected against and that the extent of negative selection scales with the severity of aneuploidy and the identity and gene abundance on the aneuploid chromosomes (*Sheltzer and Amon, 2011*). Chromosome arm fitness contribution scores ($f_c$) are taken as the chromosome arm scores derived above (section 2.1) and the sum of these scores is 1. These base values are then modified under the gene abundance model to generate a contextual fitness score ($CFS_{GA,c}$) at each time step such that…

$$CFS_{GA,c} = f_c - \frac{f_c \times |n_c - \bar{x}_p|}{\bar{x}_p}$$

$$F = \sum_{c=1}^{46} CFS_{GA,c}$$

… where $\bar{X}_p$ is the average ploidy of the population and $n_c$ is the chromosome copy number. In this model, the fitness contribution of a chromosome declines as its distance from the average ploidy increases and that the magnitude of this effect is dependent on the size of the chromosome.

## Selection on driver density

The Driver Density modality relies on assigned fitness values to chromosomes based on their relative density of tumor suppressor genes, essential genes, and oncogenes. Chromosome arm fitness contribution scores ($f_c$) are taken as the chromosome arm scores derived above (section Driver density scores) and are employed such that…

$$CFS_{TOE,c} = \frac{n_c \times TOE_c}{\bar{x}_p}$$

$$F = \sum_{c=1}^{46} CFS_{TOE,c}$$

This selection model benefits cells that have maximized the density of oncogenes and essential genes to tumor suppressors through chromosome mis-segregation.

## Hybrid selection

The hybrid model relies on selection on both gene abundance and driver densities. $CFS_{TOE,c}$ and $CFS_{GA,c}$ are both calculated and averaged such that…

$$F = \sum_{c=1}^{46} \frac{CFS_{GA,c} + CFS_{TOE,c}}{2}$$

## Neutral selection

When populations are grown under neutral selection, the fitness of each cell is constitutively set to 1 regardless of the cells' individual karyotypes.

$$F = 1$$

## Scaling selection pressure

Within each model of karyotype selection, the magnitude of selective pressure upon any karyotype, with fitness F, can be scaled by applying the scalar exponent S to produce a modified fitness score $F_M$. Thus…

$$F_M = F^S$$

For example, in the Gene Abundance model of karyotype selection, an otherwise diploid cell with three copies of chromosome 1 in a diploid population will have a F value of 0.954. Under selection-null conditions (S = 0)…

$$F_M = F^S = 0.954^0 = 1$$

… the fitness of the aneuploid cell is equivalent to that of a euploid cell. Under conditions of high selection (S = 50)…

$$F_M = F^S = 0.954^{50} = 0.097$$

…fitness of the aneuploid cell is ~10% that of the euploid cell and thus divides ~10% as frequently.

## Modeling growing and constant population dynamics

To accommodate different population size dynamics, we implemented our model using either growing, pseudo-Moran limited population dynamics and constant-size populations with approximated Wright-Fisher population dynamics.

### Simulating CIN in exponentially growing populations with pseudo-Moran limits

Populations begin with 100 founder cells with a euploid karyotype of integer value $\bar{X}_p$ and the simulation is initiated.

CFS values are calculated for each chromosome in a cell according to the chosen karyotype selection model.

Cellular fitness is calculated based on CFS values.

Selective pressure (S) is applied to fitness (F) values to modify cellular fitness ($F_M$).

Cells are checked to see if any death conditions are met and if the population limit is met. Cells die if any chromosome arm copy ($n_c$) is less than 1 or greater than 6 (unless otherwise indicated). We implemented population size limits in a pseudo-Moran fashion to reduce computational constraints. If the population size is 3000 cells or greater, a random half of the population is deleted.

Cells probabilistically divide if their fitness is greater than a random float (R) between 0 and 2. Thus...

$$R \sim U[0, 1]$$

If a cell does not divide, it restarts the cycle from CFS values are calculated for each chromosome in a cell according to the chosen karyotype selection model. If a cell divides, mis-segregations may occur.

Each copy ($n_c$) of each chromosome (c) has an opportunity to mis-segregate probabilistically. For each chromosome copy, a mis-segregation occurs if a random float (R) between 0–1 falls below $P_{misseg}$. Thus...

$$R \sim U[0, 1]$$
$$\text{Mis} - \text{segregate chromosome c if } P_{misseg,c>R}$$

If a chromosome copy is not mis-segregated, the next chromosome copy is tested. If a chromosome copy is mis-segregated, chromosome arms may be segregated separately (i.e. a reciprocal, arm-level CNA) if a random float (R) between 0 and 1 falls below $P_{break}$. Thus...

$$R \sim U[0, 1]$$
$$\text{Break chromosome c if } P_{misseg,c>R}$$

The karyotype of the cell is modified according to the results of the mis-segregation sequence above. When the mis-segregation sequence is complete, a clone of the initial cell with any reciprocal copy number alterations to its karyotype is created.

The simulation ends if it reaches 100 steps and data are exported. Otherwise, the simulation continues from CFS values are calculated for each chromosome in a cell according to the chosen karyotype selection model.

## Simulating CIN in constant-size populations with approximated Wright-Fisher dynamics

We approximated constant-size Wright-Fisher dynamics in our model by re-initiating the population at each time step and randomly drawing from the previous generation's distribution of chromosome copy numbers for each chromosome in each cell of the new population. Because the exponential pseudo-Moran model relies on proliferation rates across over-lapping generations to enact karyotype selection, such a method would not be useful here. To accommodate karyotype selection in this model, we employed an additional baseline death rate of about 20% (*Sottoriva et al., 2015*) that increases for cells with lower fitness and decreases for cells with higher fitness (see section 4.2.9). In this way, the karyotypes of the cells that die are removed from the pool of karyotypes that are drawn upon in the subsequent generation. CIN is simulated in this model as follows:

Populations begin with 4,500 founder cells and the simulation is (re-)initiated. The population begins with a euploid karyotype of integer value $\bar{X}_p$ if the population is being created for the first time.

Cells divide every step, regardless of fitness.

Chromosomes are mis-segregated in the same fashion as the exponential pseudo-Moran model above (sections 4.1.8–4.1.10).

The simulation ends if it reaches 100 steps and data are exported. Otherwise, the simulation continues from 4.2.1.

CFS values are calculated for each chromosome in a cell according to the chosen karyotype selection model.

Cellular fitness is calculated based on CFS values.

Selective pressure (S) is applied to fitness (F) values to modify cellular fitness ($F_M$).

Cells are checked to see if any death conditions are met and if the population limit is met. Cells die if any chromosome arm copy ($n_c$) is less than 1 or greater than 6 (unless otherwise indicated).

Additionally, the cells' fitness values and a random float (R) between 0 and 5 are used to determine if they die. In this way, a cell with a fitness of 1 has a 20% baseline death rate. Thus, cells die if…

$$\frac{1}{F^S + 0.001} > R \sim U[0, 5]$$

After determining cell death, the copy number distributions of each cells' chromosome arm (c) are individually stored.

The cycle repeats from 4.2.1. However, the re-initated population will have its chromosome arm copy numbers drawn from the previous generation's stored chromosome arm copy number distributions.

## Analysis of population diversity and topology in biological and simulated data

Phylogenetic trees were reconstructed from chromosome copy number profiles from live and simulated cells by calculating pairwise Euclidean distance matrices and performing complete-linkage clustering in R (*R Development Core Team, 2021*). Phylogenetic tree topology measurements were performed in R using the package phyloTop v2.1.1 (*Kendall et al., 2018*). Sackin and Colless indices of tree imbalance were calculated, normalizing to the number of tree tips. Cherry and pitchfork number were also normalized to the size of the tree. MKV is taken as the variance of individual chromosomes taken across the population, averaged across all chromosomes, then normalized to the average ploidy of the population. Average aneuploidy is calculated as the variance within a single cell's karyotype averaged across the population.

## Approximate bayesian computation

Approximate Bayesian computation was used for parameter inference of experimental data from simulated data. For this we employed the the "abc" function in the R package abc v2.1 (*Csilléry et al., 2010*). In short, a set of simulation parameters, $\theta_i$, is sampled from the prior distribution. This set of parameters corresponds to a set of simulated summary statistics, $S(y_i)$, in this case phylogenetic tree

shapes, which can be compared to the set of experimental summary statistics, $S(y_o)$. The Euclidean distance between the experimental and simulated summary statistics can then be calculated ($dS(y_i),S(y_o)$). A threshold, $T$, is then selected—0.05 in our case—which rejects the lower 1 $T$ sets of simulation parameters that correspond. The remaining parameters represent those that gave summary statistics with the highest similarity to the experimental summary statistics. These represent the posterior distribution of accepted parameters.

Bayesian model selection was performed using the "postpr" function in the same R package using tolerance threshold of 0.05 and rejection sampling method. This was used to calculate the posterior probability of each selection model *within* each growth model and the Bayes factor for each selection model with neutral selection as the null hypothesis. Bayes factors > 5 were considered substantial evidence of the alternative hypothesis.

## Sliding window analysis to tune time-steps for approximate Bayesian computation

We chose which simulation time steps to use for approximate Bayesian computation on organoid and biopsy data by repeating the inference using a sliding window of prior datasets with a width of 11 time steps (i.e. parameters from steps $\in$ [0–10], [10-20], …, [91-100]) to see if the posterior distributions would stabilize over time. We then chose simulations from 40 to 80 time steps as our prior dataset as this range provided both a stable inference and is centered around 60 time steps (analogous to 30 generations, estimated to generate a 1 cm palpable mass of ~1 billion cells).

## Cell cultivation procedures

Cal51 cells expressing stably integrated RFP-tagged histone H2B and GFP-tagged a-tubulin were generated as previously described (*Zasadil et al., 2014*). Cells were maintained at 37 °C and 5% $CO_2$ in a humidified, water-jacketed incubator and propagated in Dulbecco's Modified Eagle's Medium (DMEM) – High Glucose formulation (Cat #: 11965118) supplemented with 10% fetal bovine serum and 100 units/mL penicillin-streptomycin. Paclitaxel (Tocris Bioscience, Cat #: 1097/10) used for cell culture experiments was dissolved in DMSO. The Cal51 cells were obtained from the DSMZ-German Collection of Microorganisms and Cell Cultures and were free from mycoplasma contamination prior to study. Karyotype analysis confirms the near-diploid characteristic of the cell line and the presence of both fluorescent markers suggests they are free of other contaminating cell lines.

## Time-lapse fluorescence microscopy

Cal51 cells were transduced with lentivirus expressing mNeonGreen-tubulin-P2A-H2B-FusionRed. A monoclonal line was treated with 20 nM paclitaxel for 24, 48, or 72 hr before timelapse analysis at 37 °C and 10% $CO_2$. Five 2 µm z-plane images were acquired using a Nikon Ti-E inverted microscope with a cMos camera at 3-min intervals using a 40 X/0.75 NA objective lens and Nikon Elements software.

## Flow cytometric analysis and cell sorting

Cells were harvested with trypsin, passed through a 35 µm mesh filter, and rinsed with PBS prior to fixation in ice cold 80% methanol. Fixed cells were stored at –80 °C until analysis and sorting at which point fixed cells were resuspended in PBS containing 10 µg/ml DAPI for cell cycle analysis.

### Flow cytometric analysis

Initial DNA content and cell cycle analyses were performed on a 5 laser BD LSR II. Doublets were excluded from analysis via standard FSC/SSC gating procedures. DNA content was analyzed via DAPI excitation at 355 nm and 450/50 emission using a 410 nm long pass dichroic filter.

### Fluorescence activated cell sorting

Cell sorting was performed using the same analysis procedures described above on a BD FACS AriaII cell sorter. In general, single cells were sorted through a 130 µm low-pressure deposition nozzle into each well of a 96-well PCR plate containing 10 µl Lysis and Fragmentation Buffer cooled to 4 °C on a Eppendorf PCR plate cooler. Immediately after sorting PCR plates were centrifuged at 300 x g for 60 s. For comparison of single-cell sequencing to bulk sequencing, 1000 cells were sorted into each

'bulk' well. The index of sorted cells was retained allowing for the post hoc estimation of DNA content for each cell.

## Low-coverage single-cell whole genome sequencing

Initial library preparation for low-coverage scDNAseq was performed as previously described (*Leung et al., 2016*) and adapted for low coverage whole genome sequencing instead of high coverage targeted sequencing. Initial genome amplification was performed using the GenomePlex Single Cell Whole Genome Amplification Kit and protocol (Sigma Aldrich, Cat #: WGA4). Cells were sorted into 10 µl pre-prepared Lysis and Fragmentation buffer containing Proteinase K. DNA was fragmented to an average of 1 kb in length prior to amplification. Single cell libraries were purified on a 96-well column plate (Promega, Cat #: A2271). Library fragment distribution was assessed via agarose gel electrophoresis and concentrations were measured on a Nanodrop 2000. Sequencing libraries were prepared using the QuantaBio sparQ DNA Frag and Library Prep Kit. Amplified single-cell DNA was enzymatically fragmented to ~250 bp, 5'-phosphorylated, and 3'-dA-tailed. Custom Illumina adapters with 96 unique 8 bp P7 index barcodes were ligated to individual libraries to enable multiplexed sequencing (*Leung et al., 2016*). Barcoded libraries were amplified following size selection via AxygenAxyPrep Mag beads (Cat #: 14-223-152). Amplified library DNA concentration was quantified using the Quant-iT Broad-Range dsDNA Assay Kit (Thermo, Cat #: Q33130). Single-cell libraries were pooled to 15 nM and final concentration was measured via qPCR. Single-end 100 bp sequencing was performed on an Illumina HiSeq2500.

## Single-cell copy number sequencing data processing

Single-cell DNA sequence reads were demultiplexed using unique barcode index sequences and trimmed to remove adapter sequences. Reads were aligned to GRCh38 using Bowtie2. Aligned BAM files were then processed using Ginkgo to make binned copy number calls. Reads are aligned within 500 kb bins and estimated DNA content for each cell, obtained by flow cytometric analysis, was used to calculate bin copy numbers based on the relative ratio of reads per bin (*Garvin et al., 2015*). We modified and ran Ginkgo locally to allow for the analysis of highly variable karyotypes with low ploidy values (see Code and Data Availability). Whole-chromosome copy number calls were calculated as the modal binned copy number across an individual chromosome. Cells with fewer than 100,000 reads were filtered out to ensure accurate copy number calls (*Baslan et al., 2015*). Cells whose predicted ploidy deviated more than 32% from the observed ploidy by FACS were also filtered out. The final coverage for the filtered dataset was 0.03 (5). Single cell data extracted from *Navin et al., 2011* were separated into their individual clones and depleted of euploid cells. Single cell data from *Bolhaqueiro et al., 2019* were filtered to include only the aneuploid data that fell within the ploidies observed in the study (see Code and Data Availability).

## Review and approximation of mis-segregation rates from published Studies

We reviewed the literature to extract per chromosome rates of mis-segregation for cell lines and clinical samples. Some studies publish these rates. For those that did not, we estimated these rates by approximating the plotted incidence of segregation errors thusly:

$$\text{Approximate missegregation rate per chromosome} = \frac{\text{Observed \% frequency of errors per division}/100}{\text{Total \# modal chromosomes in sample}}$$

Modal chromosome numbers were either taken from ATCC where available or were assumed to equal 46. Observed % frequencies were approximated from published plots. Approximated rates assume that 1 chromosome is mis-segregated at a time.

## Acknowledgements

This study was supported by grants to MEB and BAW from the NCI (5R01CA234904). ARL was supported by the UW Cellular and Molecular Pathology (5 T32GM081061) and the UW Genomic Sciences Training Program (5T32HG002760) NIH training grants. NLA was supported by T32GM008692. ASZ. was supported in part by T32GM008688.Technical support comes from University of Wisconsin Carbone Cancer Center (UWCCC) Shared Resources funded by the UWCCC

Support Grant P30 CA014520 – Flow Cytometry Core Facility (1S10RR025483-01), Cancer Informatics Shared Resource, Small Molecule Screening Facility. The authors thank the UW Biotechnology Center DNA Sequencing Facility for providing Illumina sequencing services. Special thanks go to Drs. Ana Bolhaqueiro, Bas Ponsioen, and Geert Kops for the provision of scDNAseq data for our analyses and to Dr. Caitlin Pepperell for valuable comments related to approximate Bayesian computation.

## Additional information

### Competing interests

Mark E Burkard: declares the following: Medical advisory board of Strata Oncology; Research funding from Abbvie, Genentech, Puma, Arcus, Apollomics, Loxo Oncology/Lilly, and Elevation Oncology. I hold patents on microfluidic device for drug testing, and for homologous recombination and super-resolution microscopy technologies. I declare all interests without adjudicating relationship to the published work. The other authors declare that no competing interests exist.

### Funding

| Funder | Grant reference number | Author |
| --- | --- | --- |
| National Cancer Institute | R01CA234904 | Mark E Burkard |
| National Institutes of Health | R01GM141068 | Mark E Burkard |
| National Cancer Institute | P30CA014520 | Mark E Burkard |
| National Cancer Institute | F31CA254247 | Andrew R Lynch |
| National Institutes of Health | T32HG002760 | Andrew R Lynch |
| National Institutes of Health | T32GM81061 | Andrew R Lynch |
| National Institutes of Health | T32GM008692 | Nicholas L Arp |
| National Institutes of Health | T32GM008688 | Amber S Zhou |
| National Institutes of Health | T32GM140935 | Nicholas L Arp |

The funders had no role in study design, data collection and interpretation, or the decision to submit the work for publication.

### Author contributions

Andrew R Lynch, Conceptualization, Data curation, Formal analysis, Investigation, Methodology, Software; Nicholas L Arp, Formal analysis, Investigation, Writing – review and editing; Amber S Zhou, Investigation, Writing – review and editing; Beth A Weaver, Funding acquisition, Project administration, Supervision, Writing – review and editing; Mark E Burkard, Conceptualization, Funding acquisition, Methodology, Project administration, Resources, Supervision, Writing – original draft, Writing – review and editing

### Author ORCIDs

Andrew R Lynch http://orcid.org/0000-0002-0238-682X
Nicholas L Arp http://orcid.org/0000-0001-8709-0667
Beth A Weaver http://orcid.org/0000-0002-7830-3816
Mark E Burkard http://orcid.org/0000-0002-4215-7722

### Decision letter and Author response

Decision letter https://doi.org/10.7554/eLife.69799.sa1
Author response https://doi.org/10.7554/eLife.69799.sa2

## Additional files

### Supplementary files
• Transparent reporting form

### Data availability

Single-cell DNA sequencing data from this study has been deposited in NCBI SRA (PRJNA725515). All data and scripts used for modeling and analysis have been deposited in OSF at https://osf.io/snrg3/.

The following datasets were generated:

| Author(s) | Year | Dataset title | Dataset URL | Database and Identifier |
|---|---|---|---|---|
| Lynch AR, Arp NL, Zhou AS, Weaver BA, Burkard ME | 2021 | Quantifying chromosomal instability from intratumoral karyotype diversity using agent- based modeling and Bayesian inference | https://osf.io/snrg3/ | Open Science Framework, 10.17605/OSF.IO/SNRG3 |
| Lynch AR, Arp NL, Zhou AS, Weaver BA, Burkard ME | 2021 | Quantifying chromosomal instability from intratumoral karyotype diversity Quantifying chromosomal instability from intratumoral karyotype diversity | https://www.ncbi.nlm. nih.gov/bioproject/ PRJNA725515 | NCBI BioProject, PRJNA725515 |

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
