## [Editor Report]

The authors have developed a framework to quantify rates of chromosomal instability (CIN) in human tumors by fitting karyotype distributions inferred from low-depth DNA-sequencing to in silico models of CIN with karyotype selection pressures, sweeping through parameter space. This is particularly useful for the development of biomarkers for CIN, which is associated with cancer metastasis and drug resistance.

---

## [Decision Letter]

**Decision letter after peer review:**

Thank you for submitting your article "Quantifying chromosomal instability from intratumoral karyotype diversity using agent- based modeling and Bayesian inference" for consideration by *eLife*. Your article has been reviewed by 3 peer reviewers, and the evaluation has been overseen by a Reviewing Editor and Anna Akhmanova as the Senior Editor. The following individuals involved in review of your submission have agreed to reveal their identity: Trevor A Graham (Reviewer #3).

Essential revisions:

1. Modelling.

Three different models to compute cell fitness based on karyotypic alteration are explored. The construction of all these models feels a little arbitrary, and the assumptions and evolutionary dynamics in each scenario should be more comprehensively explored. Specifically:

a) In the TOE model, fitness is inversely proportional to average ploidy, so it seems higher ploidies are always selected against. Is this a reasonable assumption? Why is it necessary to divide by the average ploidy?

b) In all models, is the simulated population always "out of equilibrium". If the simulations ran for longer would an "optimal karyotype" be established. Relatedly, the dynamics appear to be strongly influenced by the copy number >6 being lethal – chromosomes (in the TOE model) which are beneficial to be gained might tend to increase copy number to 5 whereas deleterious gains reduce copy number to 1 and the population rests on that "precipice". How reasonable are these "boundary conditions" and do the dynamics change significantly if they are relaxed?

c) Gains and losses appear to be treated equivalently – again is this reasonable? Especially in the TOE model where TSG gains and OG losses (and vice versa) have differing consequences. (see also point 8 below).

The modelling assumes exponential growth to a relatively small number of cells (4500) and then randomly kills half the cells to reinitiate exponential growth from 2500 cells. This regime will influence the evolutionary dynamics of the system: the random killing will cause emerging clones to often go extinct and could exacerbate the influence of drift in the system. It also effects the influence of selection, (see for example: https://www.nature.com/articles/ng.3214). Alternative growth dynamics could be implemented – such as a Wright Fisher type model of either constant or growing populations (for the construction of a growing WF see: https://pubmed.ncbi.nlm.nih.gov/25665006/) and the influence of the growth dynamics on karyotypic heterogeneity robustly assessed.

The modelling assumes only whole chromosome missegregations (and the bioinformatics data analysis averages over sub-chromsomal sized events). The authors could consider extending their analysis to handle part-chromosome events to better represent the biological data.

2. Bayesian inference.

It is a concern that unusual prior distributions are used in the ABC inference and this effects the reliability of the inference. Figure 5F and 6C show smoothed density plots for the prior distributions – which confusingly show density for S<0 – and the true priors might instead be a series of point masses at a handful of S values. This should be clarified.

It is likely that the posterior the alteration rate and S are interrelated (high S inferred when the alteration rate is high and vice versa) – so joint posteriors should be shown. Because of the interrelationship between the parameters, it is a concern that the current parameter estimates are inaccurate – currently the prior for S has zero mass for many S values, and the inferred value of the alteration rate will depend on which values of S are explored form the prior distribution. The inference should be repeated using continuous distribution over S, a uniform distribution is suggested.

When real data is analysed, only the hybrid model is compared to data, but their Figure 3 shows the diversity depends on the underlying model of selection. The authors should implement a model selection routine (one is available with the ABC-sysbio package: https://pubmed.ncbi.nlm.nih.gov/20591907/) to test which selection paradigm (if any) best represents the data. They could also consider comparing what is arguably the "null hypothesis" of neutral evolution (S=0) to a case with selection as part of this model selection, to quantitatively determine the evidence of selection in the data.

A single threshold for acceptance in the ABC algorithm (epsilon=0.05). The authors should show this is sufficiently small: a straightforward way is to plot the mean and variance of the posteriors as a function of epsilon.

Above, the possibility of temporal effects in the model are mentioned – how do temporal ("out of equilibrium" evolutionary dynamics) affect the inference?

When public data are analysed, the growth dynamics of the breast tissue, or colorectal organoid is very unlikely to match the exponential growth assumption that is assumed by the authors. These growth dynamics likely strongly determine the pattern of observed heterogeneity (exponential growth leads to large founder effects in the extant population) and so the influence of alterative growth models needs to be explored. This can also be done within a model selection framework.

The initial state of the model used to fit to public dataset is not specified and needs to be. The authors might also consider the influence of spatial sampling confounders (the breast dataset in particular in not a well-mixed population).

3. Single cell data analysis. The data are presented as if only whole chromosomal alterations occur (e.g. figure 5B). Is this actually the case? (certainly the assumption is false in breast and colon organoid datasets) Could relative read counts in say 1MB bins for each cell be provided in the appendix to reassure the reader that the types of genetic alterations occurring in their single cell data mirror the assumptions of the modelling.

As noted above, the authors should consider modelling part-chromosome alterations. They should be able to determine in their simulations how their assumption of only whole chromosome missegregation events, despite their being part-chromosomal copy number alterations in the data, affects the accuracy of the inferred chromosomal missegregation rate.

4. The presented framework is lacking expanded characterization and validation of selection models that are biologically relevant. The current framework simply applies a scalar exponent to already published fitness models for selection. It is unclear what this exponent mirrors biologically, beyond amplifying the selection pressures already explored in existing gene abundance and driver density models.

5. Related to point (4)., how is the CIN ON-OFF model in which CIN is turned off after so many cell divisions relevant biologically? Typically, CIN is a considered a trait that evolves later in cancer progression, that once tolerated, is ongoing and facilitates development of metastasis and drug resistance. A more relevant model to explore would be that of the effect of a whole genome duplication (WGD) event on population evolution, which is thought to facilitate tolerance of ensuing missegregation events (because reduce risk of nullisomy).

5. The authors utilize two models of karyotype fitness – a gene abundance model and driver density model – to evaluate impact of specific karyotypes on cellular fitness. They also include a hybrid model whose fitness effects are simply the average of these two models, which adds little value as only a weighted average. In silico results shows inferred missegregation rates are extremely disparate across the two primary models. And while a description of these differences is provided, the presented analyses do not make clear the most important question – which of these models is more clinically relevant? Toward this, in Figure 2F, the authors claim the three models approach a triploid state – which is unsupported by the in silico results. Clearly the driver model approaches a triploid state, as previously reported. But the abundance model does not and hybrid only slightly so, given that it is simply a weighted average of these two approaches. Because the authors have developed a Bayesian strategy for inferring which model parameters best fit observed data, it would be very useful to see which model best recapitulates karyotypes observed in cancer cell lines or patient materials.

6. Topological features of phylogenetic trees, while discriminatory, are largely dependent on accurate phylogenetic tree reconstruction. The latter requires more careful consideration of cell linkages beyond computing pairwise Euclidean distances and performing complete-linkage clustering. For example, a WGD event, would appear very far from its nearest cell ancestor in Euclidean space.

7. Experimental validation of the added selection exponential factor is imperative. Works have already shown models of karyotypic evolution without additional selection exponential coefficient can accurately recover rates of missegregation observed in human cell lines and cancers by fluorescent microscopy. Incorporation of this additional weight on selection pressure has not been demonstrated or validated experimentally. This would require experimental sampling of karyotypes longitudinally and is a critical piece of this manuscript's novelty.

8. It seems like this model treats chromosome gains and losses equivalently. Is this appropriate? Chromosome loss events are much more toxic than chromosome gain events – as evidenced by the fact that haploinsufficiency is widespread, and all autosomal monosomies are embryonically-lethal while many trisomies are compatible with birth and development. Can the authors consider a model in which losses exert a more significant fitness penalty that chromosome gains? (see also point 1c above).

9. Chromosomes do not missegregate at the same rate (PMID: 29898405). This point should be discussed, and, if feasible, incorporated into the authors' models.

10. Can the authors clarify their use of live cell imaging (e.g., in Figure 6G)? Certain apparent errors that are visible by live-cell imaging (like a lagging chromosome) can be resolved correctly and result in proper segregation. Is it appropriate to directly interfer missegregation rates as is done in this paper?

11. The authors should discuss in greater detail earlier mathematical models of CIN, including PMID: 26212324, 30204765, and 12446840. How does their approach improve on this prior work?

[Editors' note: further revisions were suggested prior to acceptance, as described below.]

Thank you for resubmitting your work entitled "Quantifying chromosomal instability from intratumoral karyotype diversity using agent- based modeling and Bayesian inference" for further consideration by *eLife*. Your revised article has been evaluated by Anna Akhmanova (Senior Editor), Reviewing Editors and two of the original reviewers.

The manuscript has been improved but there are some remaining issues that need to be addressed, as outlined below:

1. The reviewers are concerned about the choice of statistics used for the ABC analysis. While these summary statistics indeed must contain some statistical signal on the rate of chromosomal instability, the current method of "phylogenetic reconstruction" which based on euclidian distances of chromosomal copy number variation is very indirect. The reviewers are not convinced that this is an optimal way to extract relevant information from the distribution of chromosomal copy number across cells. At least they are not clearly rooted in models of clonal evolution.

Why this particular set of summary statistics was chosen, how it relates to the quantity of interest (rate of chromosomal instability), how it compares to other possible ways to capture relevant information must be properly justified. The choice of these statistics should be explored in much more details in the manuscript with clear rationale given and caveats with this choice clearly stated.

2. Related to point 1 – The phylogenetic reconstruction is an unusual way to summarise the statistical data and has not been rigorously assessed. The authors should elaborate on the consequences of the possibly low accuracy of the tree reconstruction itself.

3. The manuscript is difficult to follow. For example, there is a lengthy part on validating simulations that doesn't have a very specific message and could be reduced quite a lot. Please also ensure that the manuscript narrative is accessible to the general readership of *eLife*, including defining specialist concepts, referring to figures (and figure supplements) in the order they appear in the text. Please also consider adding additional labels to the figures themselves to aid the reader and avoid over-simplification e.g. "Tet" "Dip" could be written in full, Figure 2D is missing a key etc.

4. More details are required on the ABC analysis, please address the additional comments of reviewer #3 below.

*Reviewer #3 (Recommendations for the authors):*

I focused my attention at looking at the revised ABC analysis.

I found the new analysis a little hard to follow.

It is not clear to me precisely what the "step windows" and sliding/expanding timesteps are in Fig6-S1&2. I couldn't see this explicitly explained in the methods. (I can see that the authors are trying to optimise the simulated number of cell divisions, I presume some kind of adaptive search was done)

I couldn't understand the rationale for only considering the mutation rate in these optimisations. I think that because stronger selection suppresses diversity, the strength of selection should inversely correlate with the time needed to evolve the diversity observed in the organoids.

I'm concerned about the posterior distributions shown in Figure 6-S3. In most cases the mass in the posteriors is at the extremes of the prior, which suggests that the true parameter values lie outside of the range of the prior (i.e. that the priors were too narrow). This problem appears always the case for the selection coefficient and appears to be a problem for the mutation rate in some cases too (at least it seems that the mutation rate cannot be distinguished from the boundary of 0). Possibly the same issue of a to narrow a prior applies to Figure 5F (taxol treated cell lines). The concern is that the data could be much better explained by a set of parameters located outside of the parameter space considered, which would mean that the conclusions could be incorrect.

Calculating posterior predictive distributions (analagous to Figure 6-S4&5) would help to convince the goodness of fit.

Line 479 says that a large-ish value of the acceptance threshold (epsilon=0.05) was used to retain prior data. This seems an incorrect statement: epsilon determines how many simulations are rejected but all prior data is always used. Smaller epsilons should give greater accuracy but at the cost of more computation.

---

## [Author Response]

Essential revisions:1. Modelling.Three different models to compute cell fitness based on karyotypic alteration are explored. The construction of all these models feels a little arbitrary, and the assumptions and evolutionary dynamics in each scenario should be more comprehensively explored. Specifically:a) In the TOE model, fitness is inversely proportional to average ploidy, so it seems higher ploidies are always selected against. Is this a reasonable assumption? Why is it necessary to divide by the average ploidy?

This normalization was done to account for gene balance. This fundamental premise of this model is neutral selection if all chromosomes are balanced without excess oncogenes or reduced tumor suppressors. This balance would occur equally with diploid, triploid, and tetraploid cells. By including this normalization, only imbalances which create excess oncogenes or reduce suppressors relative to the entire genome impart selection. We believe this model is most consistent with the known and specific roles of oncogenes and tumor suppressors in promoting or restraining cell proliferation.

b) In all models, is the simulated population always "out of equilibrium". If the simulations ran for longer would an "optimal karyotype" be established. Relatedly, the dynamics appear to be strongly influenced by the copy number >6 being lethal – chromosomes (in the TOE model) which are beneficial to be gained might tend to increase copy number to 5 whereas deleterious gains reduce copy number to 1 and the population rests on that "precipice". How reasonable are these "boundary conditions" and do the dynamics change significantly if they are relaxed?

We believe these boundaries are reasonable as large scale copy number alterations higher than this are rare (see PMID: 31036964 and 32054838). However, to address this, we implemented a variant of our model that considers alternative thresholds. Additionally, we agree that the CIN ON-OFF model had limited biologic relevance and removed this. To improve on this, we have changed our approach to use constant CIN for a much longer period of time (3000 time steps). We agree that WGD is a relevant phenomenon. However, others have already explicitly modeled this (see PMID: 26212324 and 32139907), so we avoid doing the same. Instead, we show that tetraploid founding cells tolerate high mis-segregation rates better than diploid founding cells.

c) Gains and losses appear to be treated equivalently – again is this reasonable? Especially in the TOE model where TSG gains and OG losses (and vice versa) have differing consequences. (see also point 8 below).

While we agree that monosomies are more detrimental than trisomies in non-cancerous tissue, this is not necessarily the case in tumors in which monosomy is often observed (see PMID: 32054838). Nevertheless, to address this critique we have now added a model variant with an additional condition in which cells experience extreme fitness penalties (90% reduction) if any chromosome is haploid. We apply this condition to all selection models and find this attenuates a ploidy increase over time in diploid cells in most selection models (see Figure 3 ‘haploid penalty’).

The modelling assumes exponential growth to a relatively small number of cells (4500) and then randomly kills half the cells to reinitiate exponential growth from 2500 cells. This regime will influence the evolutionary dynamics of the system: the random killing will cause emerging clones to often go extinct and could exacerbate the influence of drift in the system. It also effects the influence of selection, (see for example: https://www.nature.com/articles/ng.3214). Alternative growth dynamics could be implemented – such as a Wright Fisher type model of either constant or growing populations (for the construction of a growing WF see: https://pubmed.ncbi.nlm.nih.gov/25665006/) and the influence of the growth dynamics on karyotypic heterogeneity robustly assessed.

To address these concerns, we improved and more clearly detailed the prior distributions for each inference within the figure legends, we tested for karyotype convergence in each model (see Figure 3), and we demonstrate that inference under the Abundance model is robust to changes in the number of time steps included in the prior data (see Figure 6 — figure supplement 1).

The modelling assumes only whole chromosome missegregations (and the bioinformatics data analysis averages over sub-chromsomal sized events). The authors could consider extending their analysis to handle part-chromosome events to better represent the biological data.

Thank you for this critique. We have now extended the model to handle arm-level segmental aneuploidy.

2. Bayesian inference.It is a concern that unusual prior distributions are used in the ABC inference and this effects the reliability of the inference. Figure 5F and 6C show smoothed density plots for the prior distributions – which confusingly show density for S<0 – and the true priors might instead be a series of point masses at a handful of S values. This should be clarified.

We now provide data simulated using uniform distributions of mis-segregation rate and selection.

It is likely that the posterior the alteration rate and S are interrelated (high S inferred when the alteration rate is high and vice versa) – so joint posteriors should be shown. Because of the interrelationship between the parameters, it is a concern that the current parameter estimates are inaccurate – currently the prior for S has zero mass for many S values, and the inferred value of the alteration rate will depend on which values of S are explored form the prior distribution. The inference should be repeated using continuous distribution over S, a uniform distribution is suggested.

We now provide the joint posterior distributions inferred from uniform prior distributions in Figure 6C.

When real data is analysed, only the hybrid model is compared to data, but their Figure 3 shows the diversity depends on the underlying model of selection. The authors should implement a model selection routine (one is available with the ABC-sysbio package: https://pubmed.ncbi.nlm.nih.gov/20591907/) to test which selection paradigm (if any) best represents the data. They could also consider comparing what is arguably the "null hypothesis" of neutral evolution (S=0) to a case with selection as part of this model selection, to quantitatively determine the evidence of selection in the data.

We now include neutral evolution. To address these concerns, we improved and more clearly detailed the prior distributions for each inference within the figure legends, we tested for karyotype convergence in each model (see Figure 3), and we demonstrate that inference under the Abundance model is robust to changes in the number of time steps included in the prior data (see Figure 6 — figure supplement 1).

A single threshold for acceptance in the ABC algorithm (epsilon=0.05). The authors should show this is sufficiently small: a straightforward way is to plot the mean and variance of the posteriors as a function of epsilon.

This is now addressed in the new Figure 6 — figure supplement 1.

Above, the possibility of temporal effects in the model are mentioned – how do temporal ("out of equilibrium" evolutionary dynamics) affect the inference?

To address these concerns, we improved and more clearly detailed the prior distributions for each inference within the figure legends, we tested for karyotype convergence in each model (see Figure 3), and we demonstrate that inference under the Abundance model is robust to changes in the number of time steps included in the prior data (see Figure 6 — figure supplement 1).

When public data are analysed, the growth dynamics of the breast tissue, or colorectal organoid is very unlikely to match the exponential growth assumption that is assumed by the authors. These growth dynamics likely strongly determine the pattern of observed heterogeneity (exponential growth leads to large founder effects in the extant population) and so the influence of alterative growth models needs to be explored. This can also be done within a model selection framework.

We have now concurrently modeled chromosomal instability with a constant population size by approximating constant-population Wright Fisher dynamics (see Materials and methods). We find these models produce similar results at the karyotype level, addressing concerns about the effects of growth patterns on karyotype evolution in this model.

The initial state of the model used to fit to public dataset is not specified and needs to be. The authors might also consider the influence of spatial sampling confounders (the breast dataset in particular in not a well-mixed population).

We have updated the figure legends with more detailed prior and parameter settings.

3. Single cell data analysis. The data are presented as if only whole chromosomal alterations occur (e.g. figure 5B). Is this actually the case? (certainly the assumption is false in breast and colon organoid datasets) Could relative read counts in say 1MB bins for each cell be provided in the appendix to reassure the reader that the types of genetic alterations occurring in their single cell data mirror the assumptions of the modelling.

Paclitaxel causes whole-chromosome aneuploidy through chromosome mis-segregation (Scribano et al. Sci Trans Med 2021). As requested, the binned read counts are now illustrated in the new Figure 5 — figure supplement 2.

As noted above, the authors should consider modelling part-chromosome alterations. They should be able to determine in their simulations how their assumption of only whole chromosome missegregation events, despite their being part-chromosomal copy number alterations in the data, affects the accuracy of the inferred chromosomal missegregation rate.

As requested, we have now performed simulations that allow for segmental aneuploidy.

4. The presented framework is lacking expanded characterization and validation of selection models that are biologically relevant. The current framework simply applies a scalar exponent to already published fitness models for selection. It is unclear what this exponent mirrors biologically, beyond amplifying the selection pressures already explored in existing gene abundance and driver density models.

Biologically, this mirrors the extent to which aneuploid karyotypes are selected for or against, given a particular model. It is an exponent rather than a multiplier because the selection value is already transformed to a probability of division. We now provide model selection and further validation of this method.

To address this, we have greatly expanded the models and their characterization. We now explicitly include a neutral model throughout, tested various modifications of the model (Figure 3C-E), and use ABC to enable model selection (see Table 3).

We implemented cellular fitness as the sum of normalized chromosome scores such that the fitness of euploid cells is 1 and the probability of division = 0.5. In this framework, within the ‘abundance’ model, a cell with triploidy of chromosome arm 1p would have a fitness of 0.98. With no additional selection, the probability that this cell divides is 0.98 x 0.5 = 0.49.

The published fitness models for karyotype selection do not experimentally determine how fitness relates to the probability of division within a given time. For example, there is no clear reason why (or evidence indicating) an extra copy of chromosome arm 1p would reduce the probability of division from 0.5 to precisely 0.49 for a given period. The proposed model of karyotype selection that our ‘abundance’ model is based on only stipulates that aneuploidy of larger chromosomes is more detrimental than small chromosomes. Thus, these fitness values behave as arbitrary units and, therefore, we believe that adjusting and fitting an arbitrary scaling factor to the biological data is appropriate. For example, with an additional selection of S=10, the same cell with trisomy of chromosome arm 1p would divide with a probability of F^S^ x 0.5 = 0.98^10^ x 0.5 = 0.41.

We could have implemented a multiplicative framework where fitness (F_mult_) is defined as the total deviation from euploid fitness (1) multiplied by a scaling factor S (F_mult_ = S(1 - F)). For the trisomy 1p example, the same fitness value (F^S^=0.98^10^) can be achieved multiplicatively as exponentially via 1 – (9.14 x (1 - 0.98)) ~ 0.98^10^. Thus, the same fitness values can be achieved through arbitrary scaling. We regret that this may have been misinterpreted because it was implemented exponentially vs multiplicatively.

To further address this critique, we have now better fitted the S values with a flat prior probability across all values, shown how it relates to P_misseg_ in posterior probabilities (e.gs, Figure 6C, Table 3) and performed the separate analysis requested.

The selection values of F are in arbitrary units and so we believe a selection scaling factor is important to include in the model. For example, without additional selection, a hypothetical aneuploid cell with a trisomy resulting in F = 0.95 would be 5% less likely to divide than a euploid cell with F = 1. The exponent scales the selection such that when S = 2, the fitness of the trisomic cell is F ~ 0.9, or 10% less likely to divide. This scaling is necessary to enable both positive and negative selection in a system fitness is decided as the sum of chromosome scores. To further validate the additional weight on selection pressure we did the following:

1) We constrained the prior distribution of simulated data for our model selection to S=1 giving only the base fitness values without additional scaling. We, again, performed model selection on the data from Bolhaqueiro et al., 2019 and Navin et al., 2011 and found that, with this constrained prior dataset, we inferred mis-segregation rates (see Table 4) that were far below rates seen in cancer cell lines (see Figure 6E).

2) Given the initial clarification that reviewers were looking for longitudinal analysis, we leveraged data provided by the authors of Bolhaqueiro et al., 2019 where they sequenced single cells from 3 clones from organoid line 16T at 3 weeks and 21 weeks after seeding. We inferred mis-segregation rates and selective pressures in these clones at the 3-week timepoint. We did so under the Abundance model using the same prior distribution of steps given that the diversity of populations under the Abundance model rapidly reach a steady state. When we simulated additional populations using these inferred characteristics we found that the karyotype composition of the simulated populations most closely resembled the biological population than did populations simulated with the unmodified selection values (see Figure 6 — figure supplement 4). This lends credence to the biological relevance of scaled selective pressure vs. unmodified selective pressure.

We have now concurrently modeled chromosomal instability with a constant population size by approximating constant-population Wright Fisher dynamics (see Materials and Methods). We find these models produce similar results at the karyotype level, addressing concerns about the effects of growth patterns on karyotype evolution in this model.

5. Related to point (4)., how is the CIN ON-OFF model in which CIN is turned off after so many cell divisions relevant biologically? Typically, CIN is a considered a trait that evolves later in cancer progression, that once tolerated, is ongoing and facilitates development of metastasis and drug resistance. A more relevant model to explore would be that of the effect of a whole genome duplication (WGD) event on population evolution, which is thought to facilitate tolerance of ensuing missegregation events (because reduce risk of nullisomy).

We agree that this may not be relevant biologically and have removed the CIN ON-OFF scheme and updated Figure 3 to, instead, explore the convergence of karyotypes. Additionally, we agree that the CIN ON-OFF model had limited biologic relevance and removed this. To improve on this, we have changed our approach to use constant CIN for a much longer period of time (3000 time steps). We agree that WGD is a relevant phenomenon. However, others have already explicitly modeled this (see PMID: 26212324 and 32139907), so we avoid doing the same. Instead, we show that tetraploid founding cells tolerate high mis-segregation rates better than diploid founding cells.

5. The authors utilize two models of karyotype fitness – a gene abundance model and driver density model – to evaluate impact of specific karyotypes on cellular fitness. They also include a hybrid model whose fitness effects are simply the average of these two models, which adds little value as only a weighted average. In silico results shows inferred missegregation rates are extremely disparate across the two primary models. And while a description of these differences is provided, the presented analyses do not make clear the most important question – which of these models is more clinically relevant? Toward this, in Figure 2F, the authors claim the three models approach a triploid state – which is unsupported by the in silico results. Clearly the driver model approaches a triploid state, as previously reported. But the abundance model does not and hybrid only slightly so, given that it is simply a weighted average of these two approaches. Because the authors have developed a Bayesian strategy for inferring which model parameters best fit observed data, it would be very useful to see which model best recapitulates karyotypes observed in cancer cell lines or patient materials.

We agree that the CIN ON-OFF model had limited biologic relevance and removed this. To improve on this, we have changed our approach to use constant CIN for a much longer period of time (3000 time steps). We agree that WGD is a relevant phenomenon. However, others have already explicitly modeled this (see PMID: 26212324 and 32139907), so we avoid doing the same. Instead, we show that tetraploid founding cells tolerate high mis-segregation rates better than diploid founding cells.

6. Topological features of phylogenetic trees, while discriminatory, are largely dependent on accurate phylogenetic tree reconstruction. The latter requires more careful consideration of cell linkages beyond computing pairwise Euclidean distances and performing complete-linkage clustering. For example, a WGD event, would appear very far from its nearest cell ancestor in Euclidean space.

We agree that the abundance and hybrid models are unable to approach a triploid state, in earnest, as does the driver and have made that clearer in the text and improved the figure panel in question for clarity. To address your latter point on which model best fits observed data, we have implemented a model selection scheme to do this (see Table 3). This indicates the gene abundance model as the most biologically relevant and provides evidence for stabilizing selection as the primary mode of selection occurring in the organoid and biopsy data we analyzed.

7. Experimental validation of the added selection exponential factor is imperative. Works have already shown models of karyotypic evolution without additional selection exponential coefficient can accurately recover rates of missegregation observed in human cell lines and cancers by fluorescent microscopy. Incorporation of this additional weight on selection pressure has not been demonstrated or validated experimentally. This would require experimental sampling of karyotypes longitudinally and is a critical piece of this manuscript's novelty.

The selection values of F are in arbitrary units and so we believe a selection scaling factor is important to include in the model. For example, without additional selection, a hypothetical aneuploid cell with a trisomy resulting in F = 0.95 would be 5% less likely to divide than a euploid cell with F = 1. The exponent scales the selection such that when S = 2, the fitness of the trisomic cell is F ~ 0.9, or 10% less likely to divide. This scaling is necessary to enable both positive and negative selection in a system fitness is decided as the sum of chromosome scores. To further validate the additional weight on selection pressure we did the following:

1) We constrained the prior distribution of simulated data for our model selection to S=1 giving only the base fitness values without additional scaling. We, again, performed model selection on the data from Bolhaqueiro et al., 2019 and Navin et al., 2011 and found that, with this constrained prior dataset, we inferred mis-segregation rates (see Table 4) that were far below rates seen in cancer cell lines (see Figure 6E).

2) Given the initial clarification that reviewers were looking for longitudinal analysis, we leveraged data provided by the authors of Bolhaqueiro et al., 2019 where they sequenced single cells from 3 clones from organoid line 16T at 3 weeks and 21 weeks after seeding. We inferred mis-segregation rates and selective pressures in these clones at the 3-week timepoint. We did so under the Abundance model using the same prior distribution of steps given that the diversity of populations under the Abundance model rapidly reach a steady state. When we simulated additional populations using these inferred characteristics we found that the karyotype composition of the simulated populations most closely resembled the biological population than did populations simulated with the unmodified selection values (see Figure 6 — figure supplement 4). This lends credence to the biological relevance of scaled selective pressure vs. unmodified selective pressure.

8. It seems like this model treats chromosome gains and losses equivalently. Is this appropriate? Chromosome loss events are much more toxic than chromosome gain events – as evidenced by the fact that haploinsufficiency is widespread, and all autosomal monosomies are embryonically-lethal while many trisomies are compatible with birth and development. Can the authors consider a model in which losses exert a more significant fitness penalty that chromosome gains? (see also point 1c above).

While we agree that monosomies are more detrimental than trisomies in non-cancerous tissue, this is not necessarily the case in tumors in which monosomy is often observed (see PMID: 32054838). Nevertheless, to address this critique we have now added a model variant with an additional condition in which cells experience extreme fitness penalties (90% reduction) if any chromosome is haploid. We apply this condition to all selection models and find this attenuates a ploidy increase over time in diploid cells in most selection models (see Figure 3 ‘haploid penalty’).

9. Chromosomes do not missegregate at the same rate (PMID: 29898405). This point should be discussed, and, if feasible, incorporated into the authors' models.

While this may be true in some contexts, the limited data on this topic (namely Worral et al. Cell Rep. 2018 and Dumont et al. EMBO J. 2020) do not agree on which chromosomes are mis-segregated more often. Worral suggested chromosomes 1-2 are particularly mis-segregated, whereas Dumont finds chromosome 3, 6, X are the highest. These differences may be explained by a context-dependent effects that depend on the model and mechanism of mis-segregation. Worral uses nocodazole washout to generate merotelics whereas Dumont gets mis-segregation through depleting CENP-A. It is unknown which if these mechanisms, if either, is representative of the mechanisms at play in human tumors so we decided to take a general approach assuming equivalent mis-segregation rates. However, we appreciate that this will be a question for other readers and we have now added this to the discussion.

10. Can the authors clarify their use of live cell imaging (e.g., in Figure 6G)? Certain apparent errors that are visible by live-cell imaging (like a lagging chromosome) can be resolved correctly and result in proper segregation. Is it appropriate to directly interfer missegregation rates as is done in this paper?

We did not perform this live cell imaging experiment. We cite these data as being kindly offered by the Kops laboratory and they correspond to the scDNAseq data for normal colon and CRC organoids from Bolhaqueiro et al. Nat Gen. 2019. We agree that chromosome mis-segregation rates cannot be directly inferred by imaging. As you say, lagging chromosomes may resolve and segregate to the correct daughter cell. The fundamental assumption is that, although not all lagging chromosomes mis-segregate, that specimens with higher rate of lagging chromosomes have higher rates of mis-segreation. Because there is no gold-standard measure of CIN in the literature to date, we feel it is necessary to show the correlation between the two and how the data from that study relates to the inferred rates in this study. We have made this clearer in the text.

11. The authors should discuss in greater detail earlier mathematical models of CIN, including PMID: 26212324, 30204765, and 12446840. How does their approach improve on this prior work?

We now provide a more detailed discussion on prior mathematical models, incorporating these and others.

[Editors' note: further revisions were suggested prior to acceptance, as described below.]

The manuscript has been improved but there are some remaining issues that need to be addressed, as outlined below:1. The reviewers are concerned about the choice of statistics used for the ABC analysis. While these summary statistics indeed must contain some statistical signal on the rate of chromosomal instability, the current method of "phylogenetic reconstruction" which based on euclidian distances of chromosomal copy number variation is very indirect. The reviewers are not convinced that this is an optimal way to extract relevant information from the distribution of chromosomal copy number across cells. At least they are not clearly rooted in models of clonal evolution.

We agree that Euclidean distance reconstruction of CNVs is not necessarily the optimal method of phylogenetic reconstruction. The ideal approach would be to use maternal/paternal SNPs to reconstruct phylogeny. However, this is not possible with the data available from state-of-the art single-cell whole-genome sequencing (scWGS). The current state of the art for scWGS is 0.1X coverage so median sequencing coverage is only 10% of the genome. While this is amply sufficient to estimate CNVs across large genomic regions, it is not sufficient to reconstruct phylogenies by SNPs with currently available algorithms. The current work nevertheless makes a material advance over existing studies which use estimates of euclidean distances alone to infer CIN. We have expanded our discussion of this important limitation in the manuscript to acknowledge this limitation.

Why this particular set of summary statistics was chosen, how it relates to the quantity of interest (rate of chromosomal instability), how it compares to other possible ways to capture relevant information must be properly justified. The choice of these statistics should be explored in much more details in the manuscript with clear rationale given and caveats with this choice clearly stated.

Thank you for raising the question about selection of summary statistics. In Approximate Bayesian Computation (ABC), only a small number of summary statistics are employed—larger numbers impair the model (Beaumont et al. Genetics 162: 2025, 2002). To identify the optimal ones, Figure 5 Supplementary 3 evaluates the 9 possible combinations of summary statistics to infer mis-segregation rate, compared with experimental, consistent with what is thought to be the best approach (Csilléry, Katalin, et al. "Approximate Bayesian computation (ABC) in practice." Trends in ecology & evolution 25.7 (2010): 410-418.). Over 90% accuracy is achieved with 4 summary statistics—aneuploidy, MKV, Colless, and cherries—justifying the use of these statistics. This was not well described in the prior version of the manuscript—we have revised the manuscript to clearly describe the approach to select the summary statistics.

2. Related to point 1 – The phylogenetic reconstruction is an unusual way to summarise the statistical data and has not been rigorously assessed. The authors should elaborate on the consequences of the possibly low accuracy of the tree reconstruction itself.

As noted above, we have expanded the discussion about using CNVs to reconstruct phylogenetic trees, noting the possible low accuracy of this tree reconstruction.

3. The manuscript is difficult to follow. For example, there is a lengthy part on validating simulations that doesn't have a very specific message and could be reduced quite a lot. Please also ensure that the manuscript narrative is accessible to the general readership of eLife, including defining specialist concepts, referring to figures (and figure supplements) in the order they appear in the text. Please also consider adding additional labels to the figures themselves to aid the reader and avoid over-simplification e.g. "Tet" "Dip" could be written in full, Figure 2D is missing a key etc.

Thank you for this important critique. As you can see, we have substantively revised the manuscript to improve flow and remove or subsume needless detail in the text. We now ensure that figures and their supplements are in order and have improved labels.

Reviewer #3 (Recommendations for the authors):I focused my attention at looking at the revised ABC analysis.I found the new analysis a little hard to follow.It is not clear to me precisely what the "step windows" and sliding/expanding timesteps are in Fig6-S1&2. I couldn't see this explicitly explained in the methods. (I can see that the authors are trying to optimise the simulated number of cell divisions, I presume some kind of adaptive search was done)

In response to reviews, we had attempted to demonstrate that the model was robust to the number of time steps. Sliding step analysis showed that whether steps 0-10, 10-20, 20-30 etc. were included in the prior, results were similar. Expanding step analysis showed that larger numbers of steps could be included without major impact on the results. Unfortunately, this was a complex analysis and we failed to present it clearly. We have now removed the expanding step windows for simplicity and attempted to simplify the text and display/explain this analysis more clearly. We also have added a relevant section describing this analysis in the methods.

I couldn't understand the rationale for only considering the mutation rate in these optimisations. I think that because stronger selection suppresses diversity, the strength of selection should inversely correlate with the time needed to evolve the diversity observed in the organoids.

Thank you—we have now added selective pressure to these optimizations and display this in Figure 6—figure supplement 1B (to replace the expanding step windows).

I'm concerned about the posterior distributions shown in Figure 6-S3. In most cases the mass in the posteriors is at the extremes of the prior, which suggests that the true parameter values lie outside of the range of the prior (i.e. that the priors were too narrow). This problem appears always the case for the selection coefficient and appears to be a problem for the mutation rate in some cases too (at least it seems that the mutation rate cannot be distinguished from the boundary of 0). Possibly the same issue of a to narrow a prior applies to Figure 5F (taxol treated cell lines). The concern is that the data could be much better explained by a set of parameters located outside of the parameter space considered, which would mean that the conclusions could be incorrect.

We agree that the mass of posteriors should not be close to the edge of the prior. We have revised Figure 5F (now Figure 5E) by expanding the priors to higher mis-segregation rates to address this concern. The updated prior distribution is now used in the revised article. In Figure 6C, the joint inference distributions, S increases asymptotically with low mis-segregation rate. The reason this occurs is because even with infinite selective pressure, at a given rate of chromosome mis-segregation, there will remain a low number of aneuploids in the population. Although this model provides a precise estimate of mis-segregation rate (i.e. CIN), the estimates of selective pressure therefore fall in a very wide band. We infer this to mean that most aneuploid clones are adversely selected against, consistent with prior work (Lukow et al. Dev Cell 2021). We have expanded the description of this in the text.

Calculating posterior predictive distributions (analagous to Figure 6-S4&5) would help to convince the goodness of fit.

Done—see Figure 5—figure supplement 4B.

Line 479 says that a large-ish value of the acceptance threshold (epsilon=0.05) was used to retain prior data. This seems an incorrect statement: epsilon determines how many simulations are rejected but all prior data is always used. Smaller epsilons should give greater accuracy but at the cost of more computation.

Thank you—we have revised this text.